# MUC1-C intersects chronic inflammation with epigenetic reprogramming by regulating the set1a compass complex in cancer progression

Atrayee Bhattacharya[1], Atsushi Fushimi[1], Keyi Wang[1], Nami Yamashita[1], Yoshihiro Morimoto[1], Satoshi Ishikawa[1], Tatsuaki Daimon[1], Tao Liu [2], Song Liu[2], Mark D. Long [2] & Donald Kufe [1✉]

Chronic inflammation promotes epigenetic reprogramming in cancer progression by pathways that remain unclear. The oncogenic MUC1-C protein is activated by the inflammatory NF-κB pathway in cancer cells. There is no known involvement of MUC1-C in regulation of the COMPASS family of H3K4 methyltransferases. We find that MUC1-C regulates (i) bulk H3K4 methylation levels, and (ii) the COMPASS *SET1A/SETD1A* and *WDR5* genes by an NF-κB-mediated mechanism. The importance of MUC1-C in regulating the SET1A COMPASS complex is supported by the demonstration that MUC1-C and WDR5 drive expression of FOS, ATF3 and other AP-1 family members. In a feedforward loop, MUC1-C, WDR5 and AP-1 contribute to activation of genes encoding TRAF1, RELB and other effectors in the chronic NF-κB inflammatory response. We also show that MUC1-C, NF-κB, WDR5 and AP-1 are necessary for expression of the (i) KLF4 master regulator of the pluripotency network and (ii) NOTCH1 effector of stemness. In this way, MUC1-C/NF-κB complexes recruit SET1A/WDR5 and AP-1 to enhancer-like signatures in the *KLF4* and *NOTCH1* genes with increases in H3K4me3 levels, chromatin accessibility and transcription. These findings indicate that MUC1-C regulates the SET1A COMPASS complex and the induction of genes that integrate NF-κB-mediated chronic inflammation with cancer progression.

[1] Dana-Farber Cancer Institute, Harvard Medical School, Boston, MA, USA. [2] Department of Biostatistics and Bioinformatics, Roswell Park Comprehensive Cancer Center, Buffalo, NY, USA. ✉email: Donald_Kufe@dfci.harvard.edu

Polycomb group proteins form Polycomb Repressive Complex 1 (PRC1) and PRC2[1]. PRC2, which includes enhancer of zeste homolog 2 (EZH2), represses *homeobox* (*HOX*) gene expression by catalyzing H3K27 methylation[1]. PRC1 promotes gene repression by ubiquitylation of H2A[1]. PRC1/2 are counteracted by the Switch/Sucrose Non-Fermentable (SWI/SNF) family of chromatin remodeling proteins that include BAF and polybromo BAF (PBAF)[2,3]. The SWI/SNF family opposes PRC1/2-mediated repression by promoting eviction of PRC1/2 and inducing an active chromatin state[3–5]. PRC1/2 are also counteracted by the Saccharomyces cerevisiae Set 1 (COMPASS) family of H3K4 methyltransferases[3,4]. The COMPASS SET1A/B and mixed-lineage leukemia complexes activate gene expression by inducing H3K4 methylation at promoters and enhancers that competitively replace PRC1/2 repressive histone modifications[3]. SET1A/B proteins contain the SET histone methyltransferase (HMT) domain and are dependent on the WDR5, RBBP5, ASH2L and DPY30 (WRAD) core subunits for activity[3]. WDR5 is indispensable for SET1 complex assembly and HMT activity[3]. WDR5 includes a (i) WDR5-binding motif that is necessary for interacting with RBBP5 and (ii) WDR5-interacting (WIN) site that confers binding to SET1 proteins[6]. The WDR5 WIN site also associates with the H3 tail as a reader of H3K4 methylation[6]. WDR5 and ASH2L are essential for ESC self-renewal capacity[3,7,8]. By contrast, RBBP5 and DPY30 are dispensable for ESC self-renewal, but are necessary for ESC differentiation[9], in support of the importance of WRAD core proteins in dictating cell fate.

*MUC1* evolved in mammals to protect barrier tissues from loss of homeostasis[10–13]. Activation of MUC1-C in response to epithelial cell stress induces inflammatory pathways associated with wound healing[12,13]. MUC1-C binds directly to the proinflammatory NF-κB p65 transcription factor (TF) and drives NF-κB-mediated induction of *ZEB1* and the epithelial-mesenchymal transition (EMT)[12–15]. MUC1-C→NF-κB signaling also activates *EZH2* and *SUZ12*, linking EMT with PRC2-mediated epigenetic reprogramming[16,17]. In addition to activating *EZH2* expression, MUC1-C binds directly to EZH2 and promotes H3K27 methylation with repression of tumor suppressor genes[16,17]. Potential involvement of MUC1-C in counteracting PRC2-mediated gene repression has been supported by the demonstration that MUC1-C→E2F1 signaling activates the SWI/SNF embryonic stem cell BAF and PBAF chromatin remodeling complexes[18,19]. MUC1-C activates enhancer-like signatures (ELSs) in cancer stem cells (CSCs) by a BAF-mediated mechanism that increases chromatin accessibility and expression of stemness-associated genes[20]. In this way, MUC1-C activates the *NOTCH1* gene and promotes the dedifferentiation of cancer cells[18,21]. MUC1-C-induced NOTCH1 expression is associated with increases in H3K4 methylation at a proximal ELS (pELS)[20], invoking potential involvement of the COMPASS HMT family. However, unlike SWI/SNF, there is no known role for MUC1-C in the regulation of COMPASS complexes.

Triple-negative breast cancers (TNBCs) are aggressive malignancies with relatively high levels of CSCs that are functionally characterized by the capacity for self-renewal, tumorigenicity and therapeutic resistance[22]. MUC1-C drives intrinsic chronic inflammation of TNBC cells by activation of the (i) type II interferon (IFN) pathway[23], (ii) pattern recognition receptors and type I IFN pathway[24], and (iii) downstream IFN stimulated genes that promote DNA damage resistance and immune evasion[25]. MUC1-C has been implicated in driving the TNBC CSC state, but by mechanisms that remain incompletely understood[22]. The present studies focus on the involvement of MUC1-C in integrating the activation of chronic inflammation with epigenetic reprogramming as a mechanism that contributes to the CSC state

and the pathogenesis of TNBCs and potentially other cancers. NF-κB-driven signaling in cancer cells has been clearly linked to intrinsic chronic inflammation[26]. Our results demonstrate that the inflammatory MUC1-C→NF-κB pathway regulates the COMPASS *SET1A*, *WDR5* and *RBBP5* genes and H3K4 methylation in TNBC cells. We show that MUC1-C-induced SET1A/WDR5 signaling activates AP-1, which interacts with MUC1-C/NF-κB complexes in driving (i) chronic inflammatory networks, and (ii) the KLF4 pluripotency and NOTCH1 stemness factors that promote the CSC state. Our findings support a role for MUC1-C in integrating chronic inflammation with the regulation of the SET1A/WDR5 COMPASS complex and epigenetic reprogramming in TNBC progression.

## Results

**MUC1-C is necessary for SET1A expression and H3K4 methylation.** The *MUC1* gene encodes N-terminal (MUC1-N) and C-terminal (MUC1-C) subunits that form a complex at the cell membrane[11]. Activation of the complex by loss of homeostasis results in the shedding of MUC1-N from the cell surface and endocytic internalization of MUC1-C with import into the nucleus[12,13]. In this way, MUC1-C, and not MUC1-N, is expressed in the nucleus (Supplementary Fig. S1a), where it functions in the regulation of gene expression[12,13]. Analysis of BT-549 TNBC cells demonstrated that silencing MUC1-C for 7 days is associated with decreases in bulk H3K4me1 and H3K4me3 levels (Fig. 1a). The decreases in H3K4me3 were sustained over 21 days, whereas H3K4me1 levels were upregulated on days 14 and 21, which were attributed to a potential feedback mechanism (Fig. 1a). Silencing MUC1-C in MDA-MB-468 and SUM149 TNBC cells for 7 days also decreased H3K4me1 and H3K4me3 (Supplementary Fig. S1b). SET1A/B are largely responsible for genome-wide deposition of H3K4 methylation[27,28]. Accordingly, we asked if MUC1-C regulates the expression of SET1A and/or SET1B. In BT-549/tet-MUC1shRNA cells, we found that DOX-induced MUC1-C silencing downregulates expression of SET1A, but not SET1B, transcripts and protein (Fig. 1b and Supplementary Fig. S1c, d). As a control, DOX treatment of BT-549/tet-CshRNA cells had little if any effect on (i) H3K4me1 and H3K4me3 levels, and (ii) SET1A expression (Supplementary Fig. S1e). We also found that MUC1-C contributes to SET1A, and not SET1B, expression in MDA-MB-468 (Fig. 1c) and SUM149 (Supplementary Fig. S1f) cells. In addressing potential off-target effects, silencing MUC1-C in BT-549 (Fig. 1d) and MDA-MB-468 (Supplementary Fig. S1g) cells with a different MUC1shRNA#2 was associated with SET1A suppression. In support of these results, we rescued MUC1-C downregulation with expression of the MUC1-C cytoplasmic domain (tet-MUC1-C/CD), which restored SET1A expression and H3K4 methylation (Fig. 1e). Moreover, treatment with the MUC1-C inhibitor GO-203, which blocks MUC1-C homodimerization and nuclear localization[12], suppressed SET1A transcripts and protein (Fig. 1f and Supplementary Fig. S1h). As reported for TNBC cells[21,22], MUC1-C drives lineage plasticity in the dedifferentiation of castration-resistant prostate cancer (CRPC) cells[13,29]. Studies in DU-145 and LNCaP-AI CRPC cells further demonstrated that MUC1-C is necessary for bulk H3K4me3 levels (Supplementary Fig. S1i) and SET1A expression (Supplementary Fig. S1j, k). These findings indicated that MUC1-C regulates SET1A and H3K4 methylation in cancer cells.

**Targeting MUC1-C suppresses expression of the WDR5 and RBBP5 subunits.** The SET1A/B HMTs are dependent on the WDR5, RBBP5 and ASH2L proteins for catalyzing H3K4 methylation[3]. Studies in BT-549 cells demonstrated that

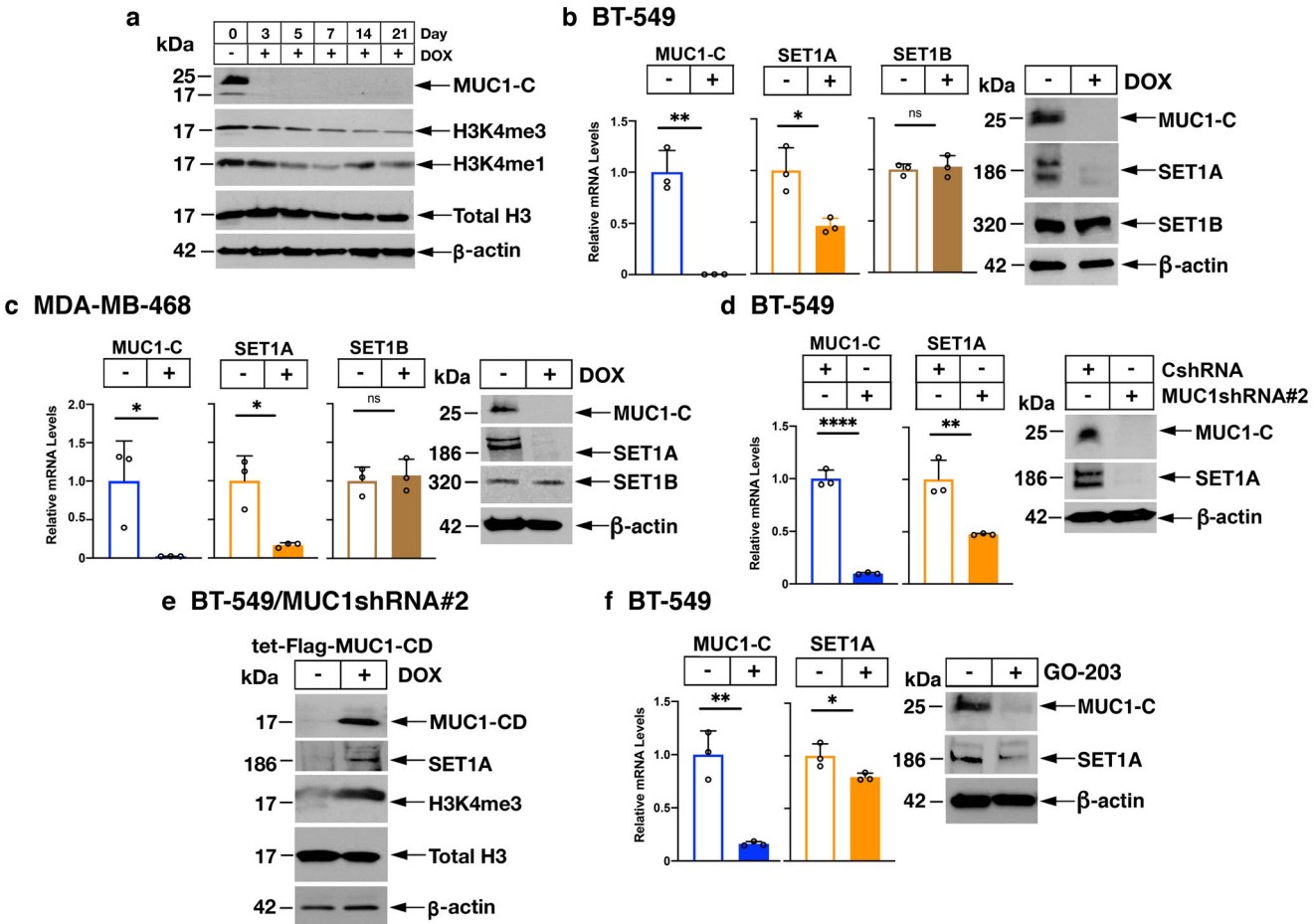

**Fig. 1 MUC1-C regulates SET1A expression and bulk H3K4me1/H3K4me3 levels. a** Lysates from BT-549/tet-MUC1shRNA cells treated with vehicle or DOX for the indicated days were immunoblotted with antibodies against the indicated proteins. **b**, **c** BT-549/tet-MUC1shRNA (**b**) and MDA-MB-468/tet-MUC1shRNA (**c**) cells treated with vehicle or DOX for 7 days were analyzed for MUC1-C, SET1A and SET1B mRNA levels by qRT-PCR. The results (mean ± SD of 3 determinations) are expressed as relative MUC1-C mRNA levels compared to that obtained for vehicle-treated cells (assigned a value of 1) (left). Lysates were immunoblotted with antibodies against the indicated proteins (right). **d** BT-549/CshRNA and BT-549/MUC1shRNA#2 cells were analyzed for MUC1-C and SET1A mRNA levels by qRT-PCR. The results (mean ± SD of 3 determinations) are expressed as relative mRNA levels compared to that obtained for CshRNA cells (assigned a value of 1) (left). Lysates were immunoblotted with antibodies against the indicated proteins (right). **e** BT-549/MUC1shRNA#2 cells transfected with a tet-Flag-MUC1-CD vector were treated with vehicle or DOX for 7 days. Lysates were immunoblotted with antibodies against the indicated proteins. **f** BT-549 cells treated with vehicle or 5 µM GO-203 for 48 h were analyzed for MUC1-C and SET1A mRNA levels by qRT-PCR. The results (mean ± SD of 3 determinations) are expressed as relative mRNA levels compared to that obtained for vehicle-treated cells (assigned a value of 1) (left). Lysates were immunoblotted with antibodies against the indicated proteins (right).

MUC1-C is necessary for the transcription of *SET1A, WDR5* and *RBBP5*, but not the *SET1B* and *ASHL2* genes (Supplementary Fig. S2a). MUC1-C was also necessary for the expression of WDR5 and RBBP5 transcripts and protein (Fig. 2a, b and Supplementary Fig. S2b, c). In contrast, like SET1B, silencing MUC1-C had no significant effect on ASH2L levels (Fig. 2a, b and Supplementary Fig. S2d). As a control, DOX treatment of BT-549/tet-CshRNA cells had no apparent effect on WDR5 and RBBP5 expression (Supplementary Fig. S2e). Similar results for WDR5 and RBBP5 were obtained in DOX-treated SUM149/tet-MUC1shRNA cells (Supplementary Fig. S2f). In focusing on WDR5 and RBBP5, silencing MUC1-C in BT-549 and MDA-MB-468 cells with MUC1shRNA#2 confirmed their down-regulation (Fig. 2c and Supplementary Fig. S2g). In addition, rescue of MUC1-C silencing in BT-549/MUC1shRNA#2 cells with MUC1-C/CD restored WDR5 and RBBP5 expression (Fig. 2d). As found for SET1A, targeting the MUC1-C cytoplasmic domain with the GO-203 inhibitor suppressed WDR5 and RBBP5 transcripts and proteins (Fig. 2e and Supplementary

Fig. S2h). These results were extended by demonstrating that MUC1-C is necessary for expression of WDR5 and RBBP5 in DU-145 (Supplementary Fig. S2i) and LNCaP-AI (Supplementary Fig. S2j) cells. BT-549 cells are dependent on MUC1-C for the CSC state, as evidenced by dedifferentiation, tumorsphere formation and tumorigenicity[21,22,30]. We found that MUC1-C, SET1A, WDR5 and RBBP5 are upregulated in BT-549 cells growing in 3D culture as mammospheres[21,22,30] (Fig. 2f). Using the same experimental conditions for BT-549 2D cells (Figs. 1b and 2a), silencing MUC1-C for 7 days in BT-549 3D mammosphere cells had similar effects on the downregulation of MUC1-C, SET1A, WDR5 and RBBP5 expression (Fig. 2g), supporting a role for MUC1-C in regulating COMPASS in enriched CSCs. SET1A catalyzes H3K4 methylation and WDR5, which is overexpressed in cancer cells[31–34], is indispensable for SET1 COMPASS HMT activity[3]. Based on those findings and the demonstration that MUC1-C has no apparent effect on SET1B and ASH2L, we focused our subsequent studies on SET1A and WDR5.

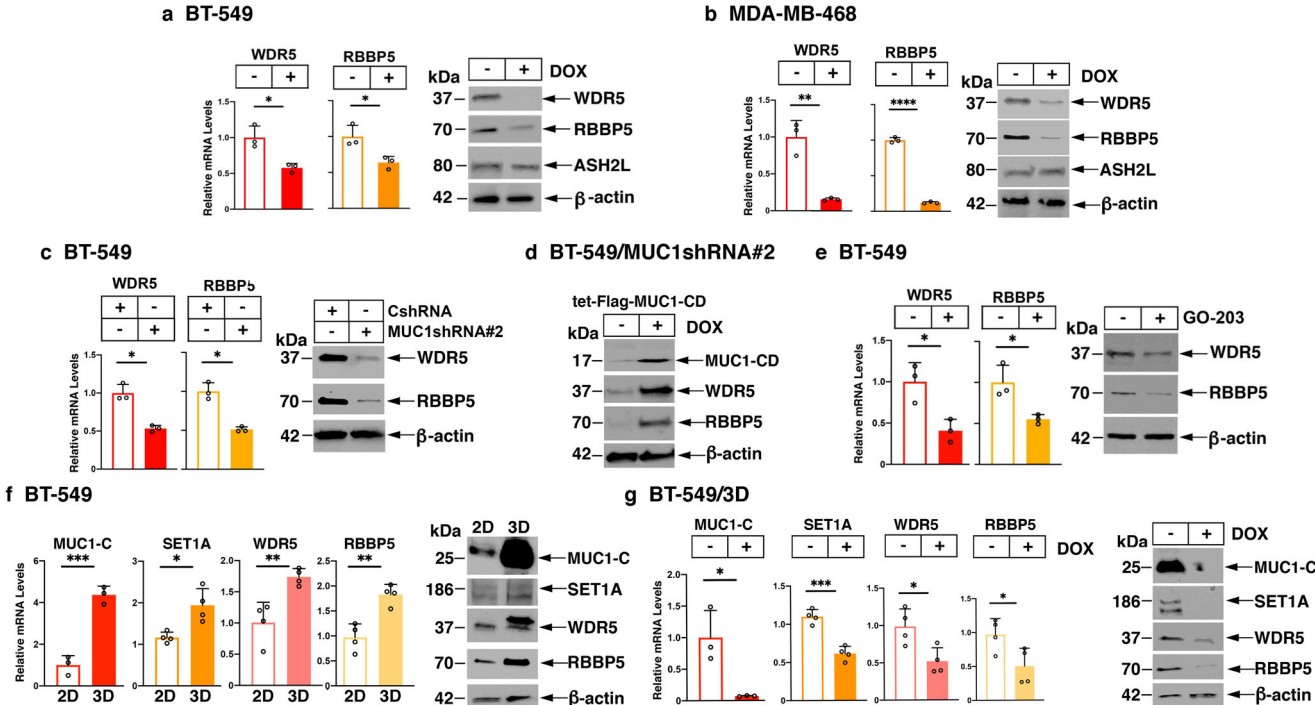

**Fig. 2 MUC1-C is necessary for WDR5 and RBBP5 expression. a**, **b** BT-549/tet-MUC1shRNA (**a**) and MDA-MB-468/tet-MUC1shRNA (**b**) cells treated with vehicle or DOX for 7 days were analyzed for WDR5 and RBBP5 mRNA levels by qRT-PCR. The results (mean ± SD of 3 determinations) are expressed as relative mRNA levels compared to that obtained for vehicle-treated cells (assigned a value of 1) (left). Lysates were immunoblotted with antibodies against the indicated proteins (right). **c** BT-549/CshRNA and BT-549/MUC1shRNA#2 cells were analyzed for WDR5 and RBBP5 mRNA levels by qRT-PCR. The results (mean ± SD of 3 determinations) are expressed as relative mRNA levels compared to that obtained for CshRNA cells (assigned a value of 1) (left). Lysates were immunoblotted with antibodies against the indicated proteins (right). **d** BT-549/MUC1shRNA#2 cells transfected with a tet-Flag-MUC1-CD vector were treated with vehicle or DOX for 7 days. Lysates were immunoblotted with antibodies against the indicated proteins. **e** BT-549 cells treated with vehicle or 5 µM GO-203 for 48 h were analyzed for WDR5 and RBBP5 mRNA levels by qRT-PCR. The results (mean ± SD of 3 determinations) are expressed as relative mRNA levels compared to that obtained for vehicle-treated cells (assigned a value of 1) (left). Lysates were immunoblotted with antibodies against the indicated proteins (right). **f** BT-549 cells grown as monolayers (2D) and as mammospheres (3D) were analyzed for expression of the indicated mRNA levels by qRT-PCR. The results (mean ± SD of 3 or more determinations) are expressed as relative mRNA levels compared to that obtained for 2D cells (assigned as value of 1)(left). Lysates were immunoblotted against the indicated proteins (right). **g** BT-549/tet-MUC1shRNA 3D mammosphere cells treated with vehicle or DOX for 7 days were analyzed for the indicated mRNA levels by qRT-PCR. The results (mean ± SD of 3 or more determinations) are expressed as relative mRNA levels compared to that obtained for vehicle-treated cells (assigned a value of 1) (left). Lysates were immunoblotted against the indicated proteins (right).

MUC1-C regulates SET1A and WDR5 by an NF-κB-mediated mechanism. MUC1-C binds directly to NF-κB p65 (RELA) and regulates the NF-κB p65 transactivation function; whereas little is known about involvement with NF-κB p50, c-REL or RELB[12–14]. MUC1-C activates PRC2 (EZH2, SUZ12 and EED) by NF-κB p65 and E2F1 signaling pathways[16,17]. We found in BT-549 cells that silencing NF-κB p65, but not E2F1, suppresses SET1A and WDR5 expression (Fig. 3a). In MDA-MB-468 cells, silencing NF-κB also downregulated SET1A and WDR5 transcripts and proteins (Fig. 3b). Comparable results were obtained in SUM149 (Supplementary Fig. S3a) and DU-145 (Supplementary Fig. S3b) cells, indicating that NF-κB drives SET1A and WDR5 in different types of cancer cells. Treatment of BT-549 (Fig. 3c) and MDA-MB-468 (Supplementary Fig. S3c) cells with the NF-κB inhibitor BAY-11 provided further support for the involvement of NF-κB signaling in driving SET1A and WDR5 expression. Analysis of the *SET1A* gene identified a putative NF-κB binding motif in a pELS (Fig. 3d). Silencing MUC1 was associated with decreases in chromatin accessibility at the pELS as evidenced by the genome browser snapshot and nuclease digestion (Fig. 3d). By contrast, there was no significant involvement of MUC1-C in regulating chromatin accessibility of the *SET1B* gene (Supplementary Fig. S3d). In further support for involvement of

MUC1-C in regulating *SET1A* expression, ChIP studies of the *SET1A* pELS demonstrated occupancy of MUC1-C and NF-κB p65 (Fig. 3e). Moreover, and consistent with the interaction between MUC1-C and NF-κB p65[14], silencing MUC1-C decreased NF-κB p65 occupancy on the *SET1A* pELS (Fig. 3e). Similar results were obtained in studies of DU-145 cells (Supplementary Fig. S3e), indicating that MUC1-C-induced activation of *SET1A* is not restricted to TNBC cells. Like *SET1A*, the *WDR5* gene contains a pELS with a putative NF-κB binding motif (Fig. 3f). We found that chromatin accessibility at the *WDR5* pELS decreases with MUC1-C silencing (Fig. 3f). We also found that the *WDR5* pELS region is occupied by MUC1-C and NF-κB p65 and that MUC1-C promotes NF-κB p65 occupancy (Fig. 3g and Supplementary Fig. S3f). As reported above, MUC1-C is necessary for RBBP5, but not ASH2L, expression. In this regard, we identified an NF-κB binding motif that is occupied by MUC1-C and NF-κB in the *RBBP5* gene (Supplementary Fig. S3g) and found that silencing NF-κB downregulates RBBP5, but not ASH2L, expression (Supplementary Fig. S3h). These findings indicated that the MUC1-C→NF-κB pathway selectively activates the *SET1A, WDR5* and *RBBP5* genes. In contrast to NF-κB, we found that E2F1 represses the expression of these genes (Supplementary Fig. S3i).

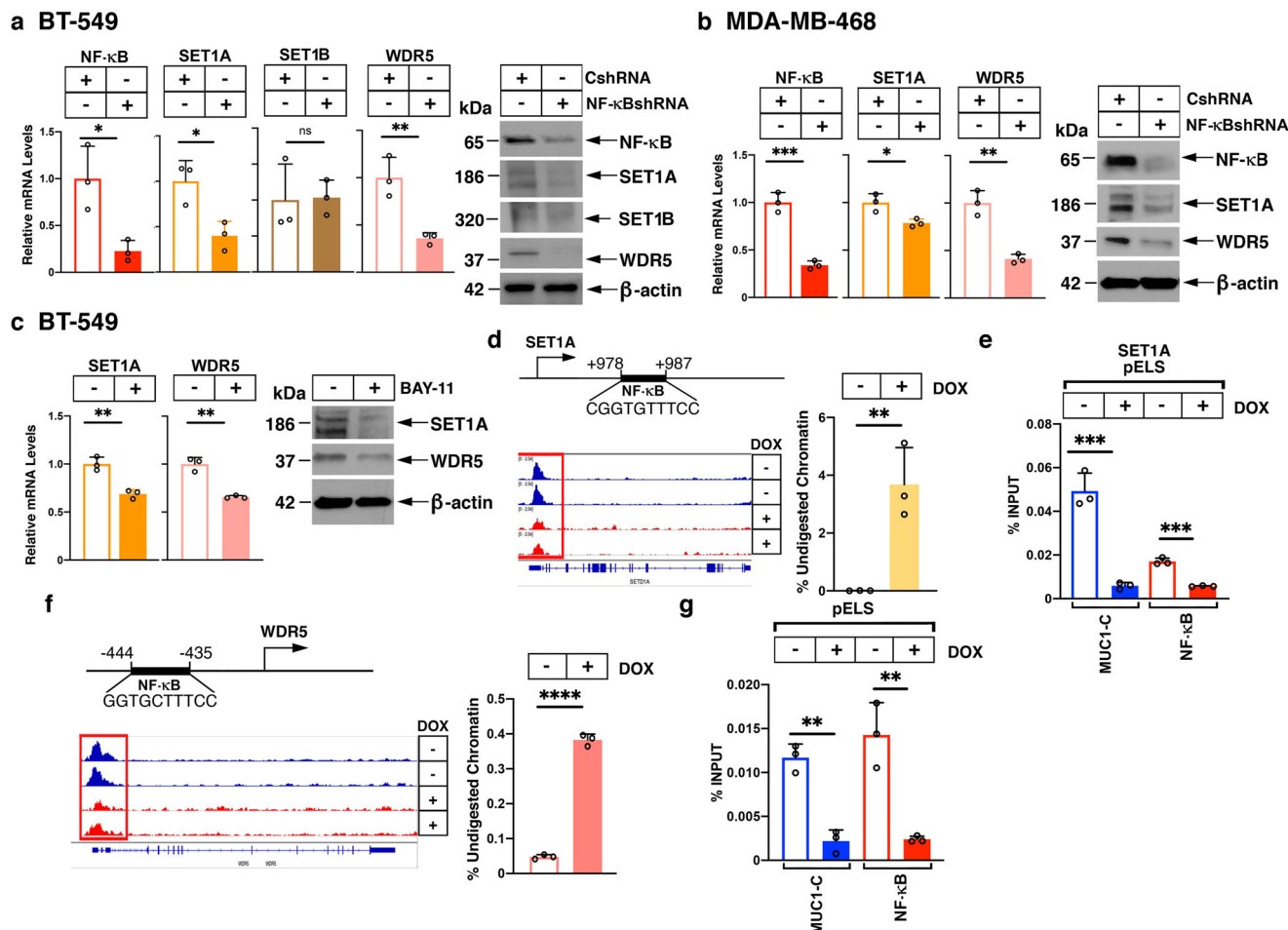

**Fig. 3 MUC1-C→NF-κB signaling activates *SET1A* and *WDR5* expression. a**, **b** BT-549 (**a**) and MDA-MB-436 (**b**) cells expressing a CshRNA or NF-κBshRNA were analyzed for NF-κB p65, SET1A and WDR5 mRNA levels by qRT-PCR. The results (mean ± SD of 3 determinations) are expressed as relative mRNA levels compared to that obtained for CshRNA cells (assigned a value of 1) (left). Lysates were immunoblotted with antibodies against the indicated proteins (right). **c** BT-549 cells treated with vehicle or 5 μM BAY-11 for 16 h were analyzed for SET1A and WDR5 mRNA levels by qRT-PCR. The results (mean ± SD of 3 determinations) are expressed as relative MUC1-C mRNA levels compared to that obtained for vehicle-treated cells (assigned a value of 1) (left). Lysates were immunoblotted with antibodies against the indicated proteins (right). **d** Scheme of *SET1A* with highlighting of a pELS containing an NF-κB binding motif. Genome browser snapshot of ATAC-seq data from the *SET1A* pELS region in BT-549/tet-MUC1shRNA cells treated with vehicle or DOX for 7 days (left). Chromatin was analyzed for accessibility by nuclease digestion (right). The results are expressed as % undigested chromatin (mean ± SD and individual values). **e** Soluble chromatin from BT-549/tet-MUC1shRNA cells treated with vehicle or DOX for 7 days was precipitated with anti-MUC1-C and anti-NF-κB p65 (right). The DNA samples were amplified by qPCR with primers for the *SET1A* pELS region. The results (mean ± SD of 3 determinations) are expressed as percent input. **f** Scheme of *WDR5* with highlighting of a pELS containing the indicated NF-κB binding motif. Genome browser snapshot of ATAC-seq data from the *WDR5* pELS in BT-549/tet-MUC1shRNA cells treated with vehicle or DOX for 7 days (left). Chromatin was analyzed for accessibility by nuclease digestion (right). The results are expressed as % undigested chromatin (mean ± SD and individual values). **g** Soluble chromatin from BT-549/tet-MUC1shRNA cells treated with vehicle or DOX for 7 days was precipitated with anti-MUC1-C and anti-NF-κB p65 (right). The DNA samples were amplified by qPCR with primers for the *WDR5* pELS region. The results (mean ± SD of 3 determinations) are expressed as percent input.

**MUC1-C and WDR5 regulate the expression of genes encoding the AP-1 family.** In exploring whether MUC1-C interacts with the SET1A complex, we found from analysis of nuclear lysates that MUC1-C co-immunoprecipitates with SET1A and WDR5 (Fig. 4a). These results suggested that MUC1-C may play a direct role in regulating genes that are activated by the SET1A/WDR5 complex. In addressing this notion, we analyzed RNA-seq data from BT-549 cells with DOX-inducible MUC1-C and WDR5 silencing to identify shared sets of suppressed and activated genes (log2FC.1; padjust = 0.05) (Fig. 4b). As demonstrated above, silencing MUC1-C suppressed *SET1A, WDR5* and *RBBP5* transcription (Supplementary Fig. S2a); however, downregulation of their expression, as determined by RNA-seq, did not achieve significance using DESeq2 (Fig. 4b). In searching for common

MUC1- and WDR5-regulated genes, we identified 100 and 24 that are repressed and activated, respectively (Fig. 4c). Gene Set Enrichment Analysis (GSEA) demonstrated 22 common pathways that are significantly downregulated by MUC1 and WDR5 silencing (Supplementary Fig. S4a–c). Further analysis identified that, like MUC1, WDR5 significantly associates with activation of the GO DNA BINDING TRANSCRIPTION FACTOR ACTIVITY (Fig. 4d and Supplementary Fig. S4d) and TRANSCRIPTION REGULATOR ACTIVITY (Supplementary Fig. S4e, f) gene signatures[23–25]. A comprehensive genome-wide analysis of RNA-seq and ATAC-seq data demonstrated that MUC1-C-induced differentially accessible regions align with genes regulated by the AP-1 family[20]. Of interest in this regard, analysis of MUC1-C and WDR5-activated genes identified those

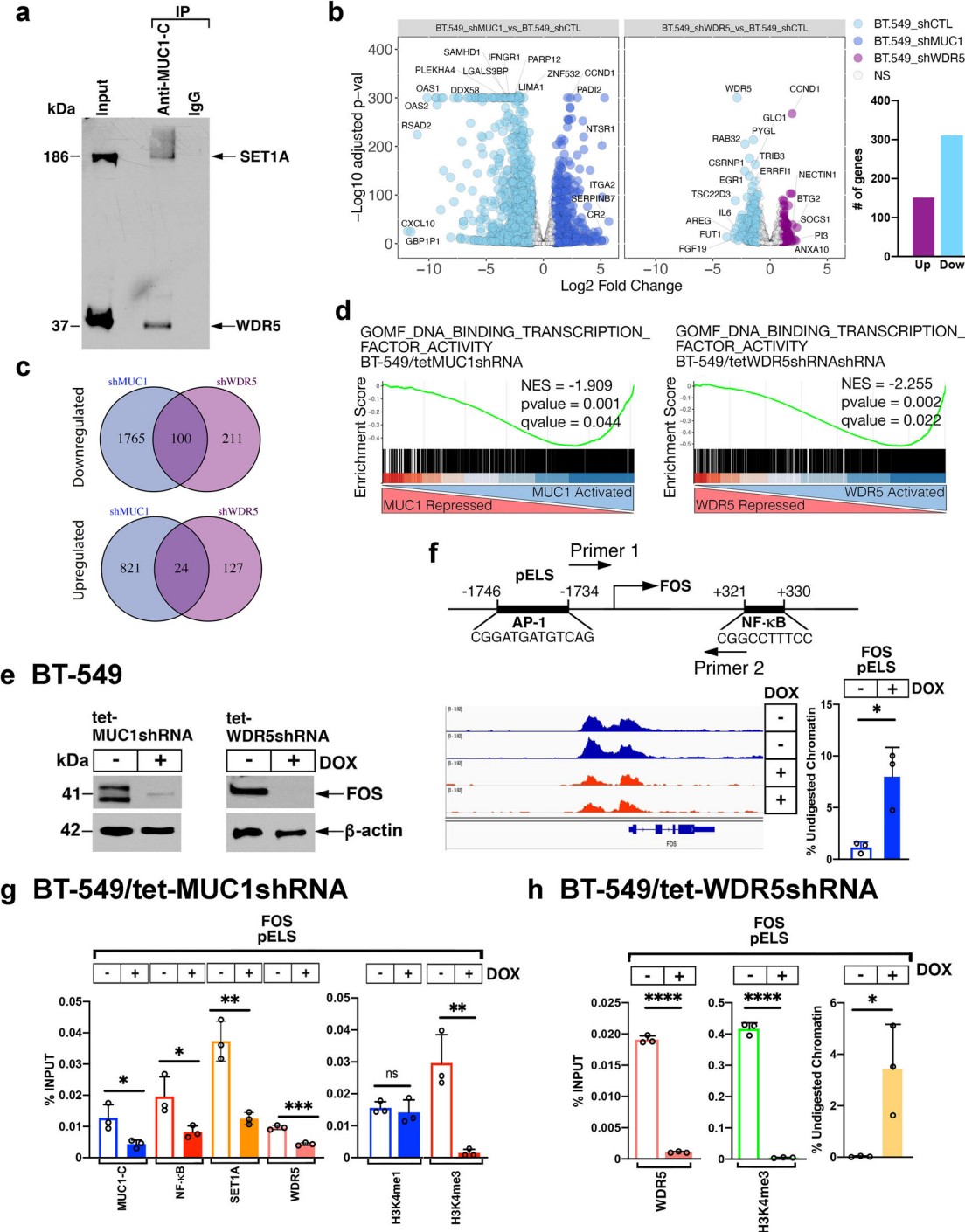

**Fig. 4 MUC1-C and WDR5 are necessary for the activation of the *FOS* gene. a** Nuclear lysates from BT-549 cells were immunoprecipitated with anti-MUC1-C or a control IgG. Input proteins and the precipitates were immunoblotted with antibodies against the indicated proteins. **b** RNA-seq was performed in triplicates on BT-549 cells silenced for MUC1 and WDR5. The datasets were analyzed for effects of MUC1-C and WDR5 silencing on repressed and activated genes as depicted by the volcano plots (left) and barplot (right). **c** Venn Diagram depicting the overlap of 100 downregulated genes and 24 upregulated genes in BT-549 cells silenced for MUC1 and WDR5. **d** GSEA of the MUC1 (left) and WDR5 (right) RNA-seq datasets using the GOMF DNA BINDING TRANSCRIPTION FACTOR ACTIVITY gene signature. **e** Lysates from BT-549/tet-MUC1shRNA (left) and BT-549/tet-WDR5shRNA cells (right) treated with vehicle or DOX for 7 days were immunoblotted with antibodies against the indicated proteins. **f** Genome browser snapshot of ATAC-seq data from the *FOS* pELS region in BT-549/tet-MUC1shRNA cells treated with vehicle or DOX for 7 days (left). Chromatin was analyzed for accessibility of the pELS by nuclease digestion (right). The results are expressed as % undigested chromatin (mean ± SD and individual values). **g** BT-549/tet-MUC1shRNA cells were treated with vehicle or DOX for 7 days. Soluble chromatin was precipitated with a control IgG, anti-MUC1-C, anti-NF-κB, anti-SET1A and anti-WDR5 (left) or a control IgG and anti-H3K4me3 (right). **h** BT-549/tet-WDR5shRNA cells were treated with vehicle or DOX for 7 days. Soluble chromatin was precipitated with anti-WDR5 and anti-H3K4me3 (left). The DNA samples were amplified by qPCR with primers for the *FOS* pELS. The results (mean ± SD of 3 determinations) are expressed as a percentage of the input DNA for each sample. Chromatin was analyzed for accessibility of the *FOS* pELS by nuclease digestion (right). The results (mean ± SD of 3 determinations) are expressed as % undigested chromatin.

encoding members of the AP-1 family (Supplementary Fig. S4g). We therefore confirmed by qRT-PCR that MUC1 and WDR5 are necessary for the expression of FOS, JUNB, JUND, JDP2, ATF3 and BATF3; whereas silencing MUC1-C and WDR5 had little if any effect on JUN transcripts (Supplementary Fig. S4h). By extension, MUC1-C and WDR5 were necessary for the expression of FOS (Fig. 4e), and to a lesser extent the JUNB, JUND and JDP2 proteins (Supplementary Fig. S4i). We therefore first focused on FOS, which is induced by stress, forms heterodimers with JUN, and is required for inflammatory memory[35–38]. In investigating the basis for FOS activation, we identified a pELS that includes NF-κB and AP-1 binding motifs and responds to MUC1-C silencing with loss of chromatin accessibility (Fig. 4f)[20]. We also found that the pELS is occupied by MUC1-C, NF-κB, SET1A and WDR5 and that silencing MUC1-C decreases their occupancy, as well as the H3K4me3 mark (Fig. 4g). Analysis of the FOS pELS further demonstrated that WDR5 is necessary for the deposition of H3K4me3 and the opening of chromatin (Fig. 4h).

To extend these results, we next focused on the ATF3 gene, which is activated by inflammation, interacts with NF-κB and promotes the progression of TNBC and other types of cancer[39–41]. Activating TFs, such as ATF3, heterodimerize with JUN and are also capable of binding to DNA in the absence of JUN[42,43]. Analysis of ATF3 demonstrated that silencing MUC1-C decreases chromatin accessibility of dELS and pELS regions, which include NF-κB motifs (Supplementary Fig. S5a). Immunoblot analysis indicated that MUC1-C and WDR5 are necessary for full-length ATF3 protein, but not JUN, expression (Supplementary Fig. S5b). Accordingly, we focused on the ATF3 dELS and identified occupancy by MUC1-C, NF-κB, SET1A and WDR5, which was decreased by MUC1-C silencing (Supplementary Fig. S5c). MUC1-C silencing also decreased H3K4me3 deposition (Supplementary Fig. S5d). Further analysis of the ATF3 dELS demonstrated that WDR5 is necessary for the H3K4me3 mark and chromatin accessibility (Supplementary Fig. S5e).

**MUC1-C forms a nuclear complex with SET1A, WDR5 and AP-1 that drives an NF-κB inflammatory regulatory network**. AP-1 associates with NF-κB p65 and regulates the NF-κB transactivation function[37,39–41]. We found that, like MUC1-C and NF-κB, the AP-1 JUN, FOS and ATF3 proteins occupy the SET1A and WDR5 pELSs and that silencing MUC1-C decreases their occupancy (Fig. 5a). Comparable results were obtained by silencing NF-κB (Fig. 5b), indicating that MUC1-C increases FOS and ATF3 expression in a positive feedback loop with the inflammatory NF-κB network. GSEA further demonstrated that, like MUC1, WDR5 significantly associates with activation of the HALLMARK TNFA SIGNALING VIA NFKB (Fig. 5c) and HALLMARK INFLAMMATORY RESPONSE (Supplementary Fig. S6a) gene signatures. From these analyses, we identified an overlap of MUC1-C- and WDR5-induced genes encoding (i) TNF pathway effectors, such as TRAF1[44], TNFAIP2[45] and TNFAIP6[46], (ii) members of the NF-κB family, including NFKB1[47] and RELB[48] and (iii) the immunosuppressive IL-6[49] and PTGS2[50] factors (Supplementary Fig. S6b–d) that collectively contribute to chronic inflammation. These findings were of interest in that, to our knowledge and unlike MUC1-C[12], WDR5 has not been previously linked to intrinsic cancer cell inflammatory signaling. In extending these results with more detailed studies of the TRAF1 gene, we identified a pELS region with NF-κB and AP-1 binding motifs upstream to the TSS (Fig. 5d). Silencing MUC1-C was associated with marked decreases in chromatin accessibility of the pELS as evidenced by the genome

browser snapshot and nuclease digestion (Fig. 5d)[20]. Silencing MUC1-C was also associated with decreases in NF-κB, SET1A and WDR5, as well as JUN, FOS and ATF3, occupancy, and decreases in the H3K4me3 mark (Fig. 5e). In addition, silencing JUN decreased SET1A, WDR5, FOS and ATF3 occupancy, and the H3K4me3 mark (Fig. 5f). Similar effects of MUC1-C silencing were found from analysis of the RELB promoter region; that is, decreases in (i) chromatin accessibility of a pELS (Supplementary Fig. S7a), (ii) occupancy by MUC1-C, NF-κB, SET1A, WDR5, JUN, FOS and ATF3 (Supplementary Fig. S7b) and (iii) the H3K4me3 mark (Supplementary Fig. S7c). These findings provided support for a model in which the MUC1-C→SET1A/WDR5 pathway plays a role in regulating activation of the TRAF1 and RELB genes that promote TNF- and NF-κB-mediated chronic inflammation and memory.

**MUC1-C→WDR5→AP-1 signaling induces the KLF4 pluripotency factor and NOTCH1 stemness-associated genes**. MUC1-C-driven chronic inflammation promotes cancer progression[12,13,51]. Whereas the above results support the involvement of the SET1A/WDR5 complex in an NF-κB inflammatory network, to our knowledge, there is no reported association of COMPASS with stemness. In this context, we found by analyzing the BENPORATH ES_1[52] and MALTA CURATED STEMNESS MARKERS gene signatures that MUC1-C and WDR5 are necessary for expression of pluripotency factor and stemness-associated genes, which included SALL4, HIF1A, BMI1, LGR5, KLF4 and NOTCH1 (Fig. 6a, b). We confirmed by qRT-PCR that MUC1 and WDR5 increase expression of (i) SALL4, an ESC effector of pluripotency and self-renewal[53], (ii) HIF1A, which is overexpressed in human cancers in association with hypoxia[54], (iii) BMI1, a component of PRC1 and effector of H2A ubiquitylation that binds directly to MUC1-C[55], and (iv) LGR5, which is necessary for maintenance of breast CSCs[56] (Fig. 6c, d). Nonetheless, these results do not exclude the possibility that, in addition to WDR5, MUC1-C regulates other effectors of SALL4, HIF1A, BMI1 and LGR5 expression. MUC1-C induces the OCT4, SOX2, KLF4, MYC (OSKM) pluripotency factors in TNBC and other cancer cells[12,29,30,51,57]. Of interest in this regard, WDR5 is necessary for efficient OSKM-mediated somatic cell reprogramming[7] but is not known to induce OSKM expression. Among the OSKM factors, KLF4 functions as a master regulator of pluripotency enhancer networks[58]. In investigating the above observation that MUC1-C and WDR5 associate with KLF4 expression, we identified a KLF4 pELS that includes NF-κB and AP-1 binding motifs and found that silencing MUC1-C suppresses chromatin accessibility of that region (Fig. 6e). We found that occupancy of the KLF4 pELS by MUC1-C, SET1A, WDR5, JUN, FOS and ATF3 is decreased by silencing MUC1-C (Fig. 6f) and JUN (Fig. 6g). Silencing MUC1-C, WDR5 and JUN also decreased (i) the H3K4me3 mark on the KLF4 pELS (Fig. 6h) and (ii) KLF4 expression (Supplementary Fig. S8a, b).

The NOTCH1 gene, which is linked to stemness in cancer cells[20,59,60], includes a pELS (−1575 to −1590 bp upstream to the TSS) that contains AP-1 and NF-κB binding motifs (Fig. 7a)[20]. MUC1-C promotes JUN/AP-1 and BAF occupancy of the NOTCH1 pELS in association with increases in chromatin accessibility, H3K4me3 levels and NOTCH1 expression[20]. Here, we detected occupancy of NF-κB, SET1A and WDR5, as well as JUN, FOS and ATF3, on the NOTCH1 pELS that was decreased by MUC1-C silencing (Fig. 7b). In addition, silencing WDR5 decreased the H3K4me3 mark and chromatin accessibility at the NOTCH1 pELS in association with downregulation of NOTCH1 expression (Fig. 7c). Along these lines, we found that JUN is necessary for SET1A, WDR5, FOS and ATF3 occupancy, as well

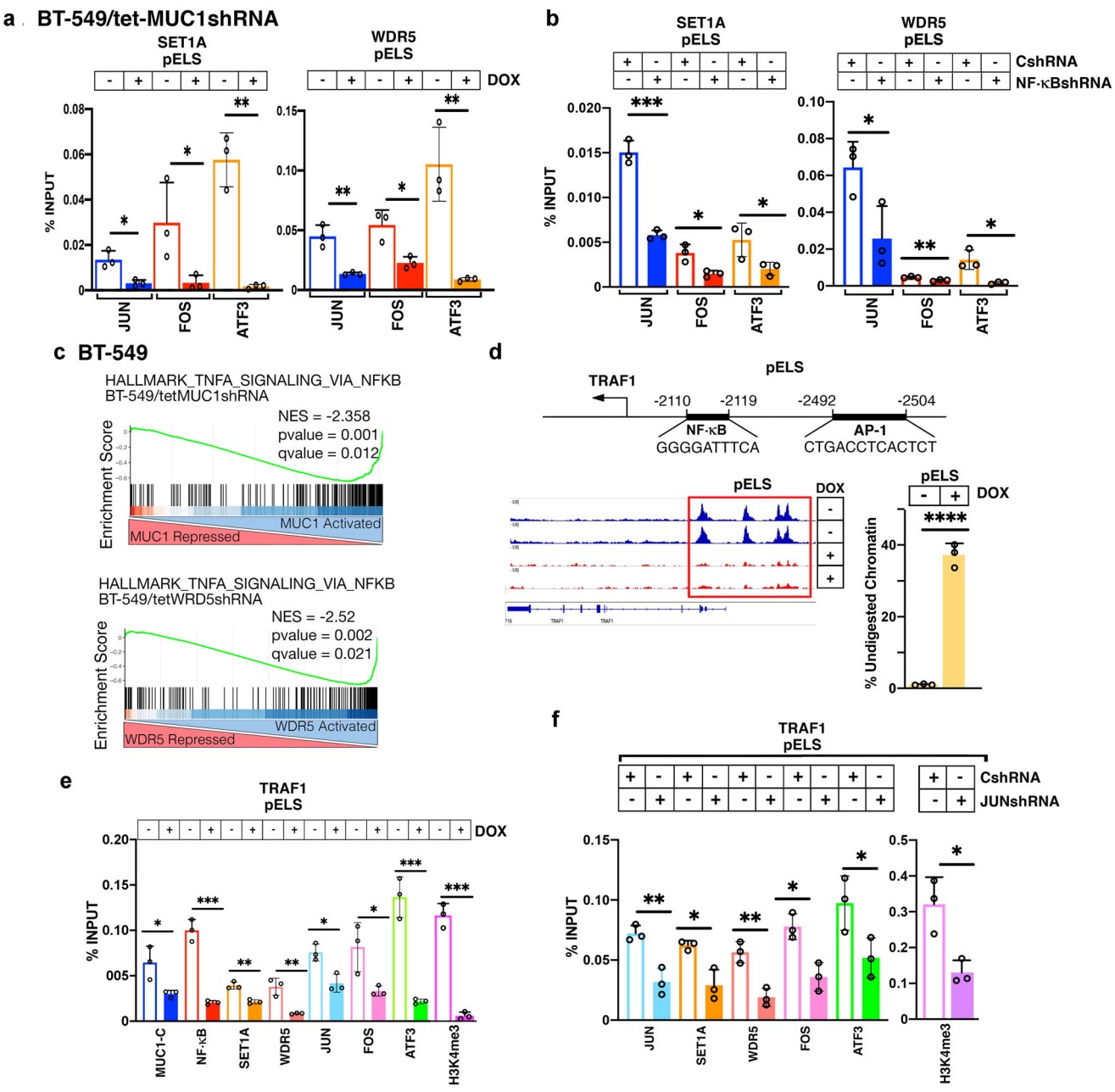

**Fig. 5 MUC1-C and WDR5 regulate common sets of genes involved in chronic activation of the NF-κB inflammatory response. a, b** Soluble chromatin from (i) BT-549/tet-MUC1shRNA cells treated with vehicle of DOX for 7 days (**a**) and (ii) BT-549/CshRNA and BT-549/NF-κBshRNA cells (**b**) was precipitated with a control IgG, anti-JUN, anti-FOS and anti-ATF3. The DNA samples were amplified by qPCR with primers for the *SET1A* (left) and *WDR5* (right) pELS regions. The results (mean ± SD of 3 determinations) are expressed as percent input. **c** GSEA of the RNA-seq datasets from MUC1-C- (left) and WDR5- (right) silenced cells using the HALLMARK TNFA SIGNALING VIA NFKB gene signature. **d** Schema of the *TRAF1* gene with highlighting of pELS region that includes NF-κB and AP-1 motifs. Genome browser snapshot of ATAC-seq data from the *TRAF1* pELS region in BT-549/tet-MUC1shRNA cells treated with vehicle or DOX for 7 days (left). Chromatin was analyzed for accessibility by nuclease digestion (right). The results are expressed as % undigested chromatin (mean ± SD and individual values). **e** Soluble chromatin from BT-549/tet-MUC1shRNA cells treated with vehicle or DOX for 7 days was precipitated with antibodies against the indicated proteins. The DNA samples were amplified by qPCR with primers for the *TRAF1* pELS regions. The results (mean ± SD of 3 determinations) are expressed as percent input. **f** Soluble chromatin from BT-549/CshRNA and BT-549/JUNshRNA cells was precipitated with the indicated proteins. The DNA samples were amplified by qPCR with primers for the *TRAF1* pELS. The results (mean ± SD of 3 determinations) are expressed as percent input.

as the H3K4me3 mark (Fig. 7d). In concert with these results, silencing WDR5 and thereby NOTCH1 was associated with loss of proliferative capacity (Fig. 7e) and, importantly, self-renewal as evidenced by suppression of tumorsphere formation (Fig. 7f).

Association of MUC1, WDR5 and SET1A expression in TNBC tumors with poor clinical outcomes. Of potential clinical relevance for the MUC1-C→WDR5 pathway, GEPIA2 analysis[61] of 135 TNBC tumors in the TCGA-BRCA dataset demonstrated significant upregulation of MUC1 and WDR5, but not SET1A, expression compared to normal breast tissue (Fig. 8a). Patients with MUC1-high vs MUC1-low TNBC tumors had significant decreases in relapse-free survival

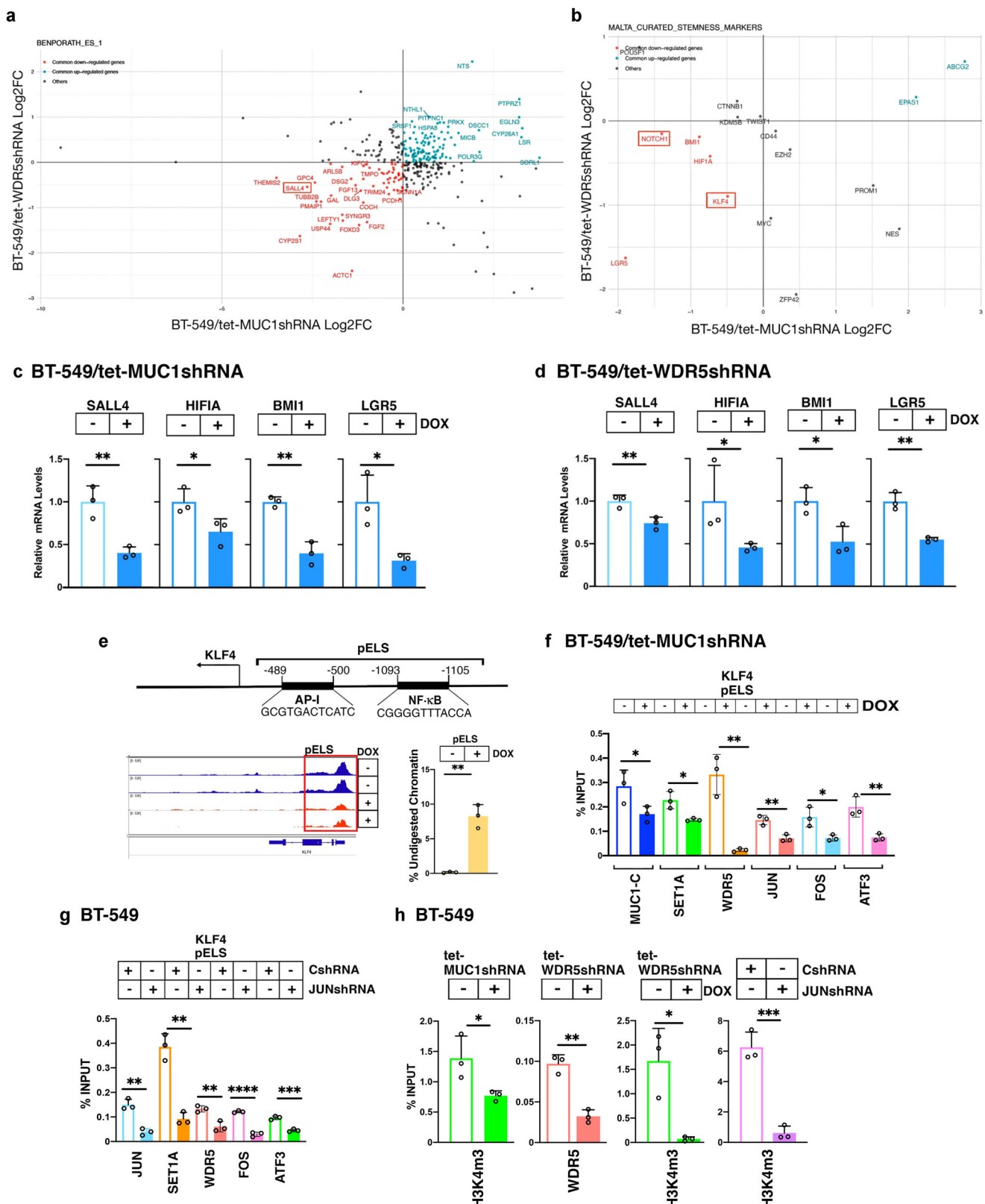

(Fig. 8b). Moreover, patients with high vs low expression of WDR5 (Fig. 8c), as well as SET1A (Fig. 8d), experienced significantly poorer clinical outcomes. These findings provided further support for the involvement of MUC1-C-induced activation of the COMPASS SET1A/WDR5 complex in promoting TNBC progression (Fig. 8e).

## Discussion

TrxG proteins form the SWI/SNF chromatin remodeling and COMPASS HMT complexes[3]. The SWI/SNF family includes BAF and PBAF[2,3]. MUC1-C induces expression of the BAF and PBAF subunits by E2F1-mediated transactivation[18,19]. However, there has been no known involvement of MUC1-C in the regulation of

**Fig. 6 MUC1-C and WDR5 are necessary for the activation of the *KLF4* pluripotency gene. a, b** Overlap of activated and repressed genes in MUC1- and WDR5-silenced cells using the BENPORATH ES 1 (**a**) and MALTA CURATED STEMNESS MARKERS (**b**) gene signatures. **c, d** BT-549/tet-MUC1shRNA (**c**) and BT-549/tet-WDR5shRNA (**d**) cells treated with vehicle or DOX for 7 days were analyzed for the indicated mRNA levels by qRT-PCR. The results (mean ± SD of 3 determinations) are expressed as relative mRNA levels compared to that obtained for vehicle-treated cells (assigned a value of 1). **e** Schema of *KLF4* with highlighting of a pELS containing the indicated NF-κB and AP-1 binding motifs. Genome browser snapshot of ATAC-seq data from the *KLF4* pELS in BT-549/tet-MUC1shRNA cells treated with vehicle or DOX for 7 days (left). Chromatin was analyzed for accessibility by nuclease digestion (right). The results are expressed as % undigested chromatin (mean ± SD and individual values). **f** Soluble chromatin from BT-549/tet-MUC1shRNA cells treated with vehicle or DOX for 7 days cells was precipitated with antibodies against the indicated proteins. **g** Soluble chromatin from BT-549/CshRNA and BT-549/JUNshRNA cells was precipitated with antibodies against the indicated proteins. **h** Soluble chromatin from (i) BT-549/tet-MUC1shRNA and BT-549/tet-WDR5shRNA cells treated with vehicle or DOX for 7 days cells (left, middle panels) and (ii) BT-549/CshRNA and BT-549/JUNshRNA (right panel) was precipitated with antibodies against the indicated proteins. The DNA samples were amplified by qPCR with primers for the *KLF4* pELS. The results (mean ± SD of 3 determinations) are expressed as percent input.

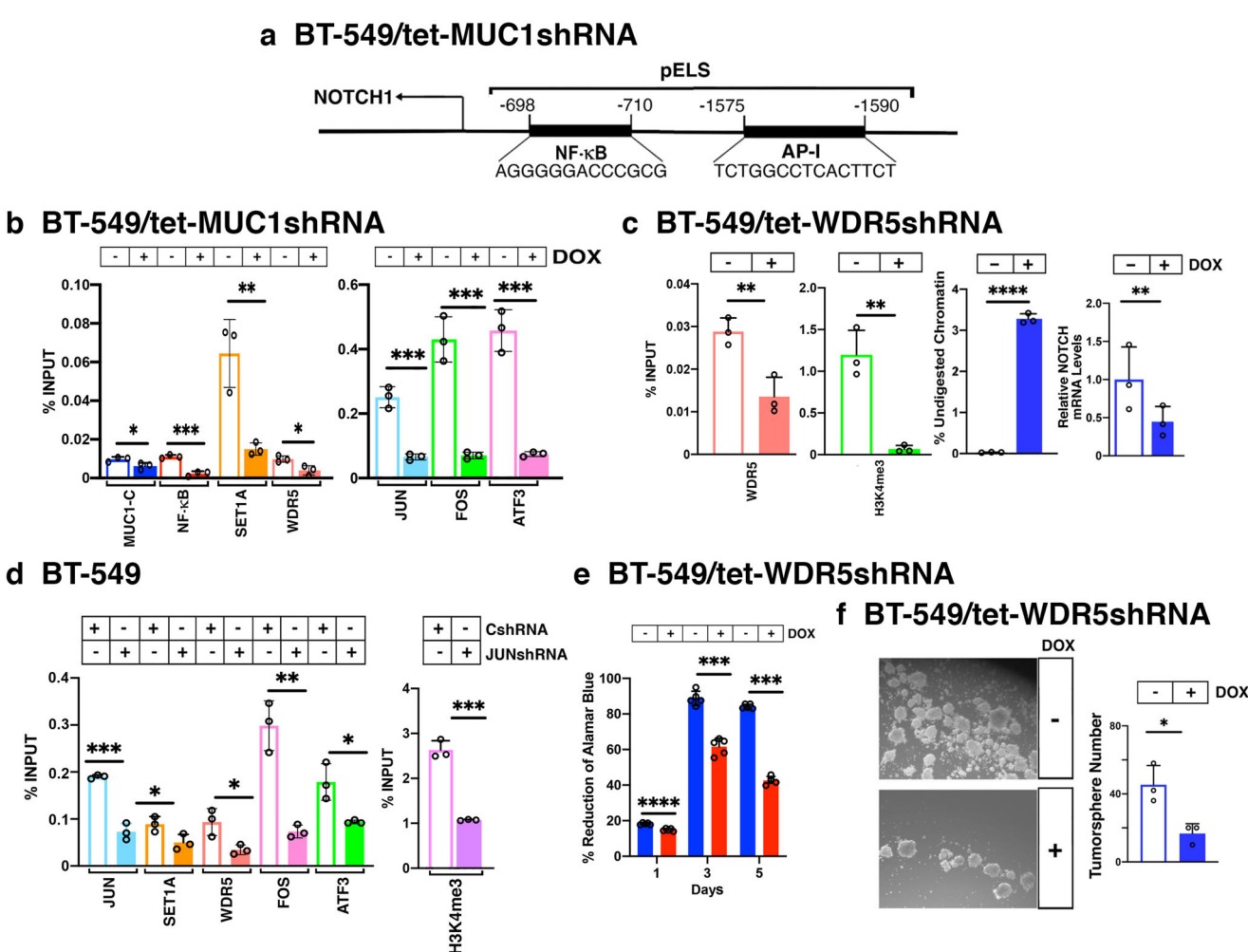

**Fig. 7 MUC1-C and WDR5 induce the *NOTCH1* gene and self-renewal capacity. a** Schema of *NOTCH1* highlighting localization of the pELS. **b** Soluble chromatin from BT-549/tet-MUC1shRNA cells treated with vehicle or DOX for 7 days was precipitated with antibodies against the indicated proteins. The DNA samples were amplified by qPCR with primers for the *NOTCH1* pELS. The results (mean ± SD of 3 determinations) are expressed as percent input. **c** BT-549/tet-WDR5shRNA cells were treated with vehicle or DOX for 7 days. Soluble chromatin was precipitated with anti-WDR5 and anti-H3K4me3. The DNA samples were amplified by qPCR with primers for the *NOTCH1* pELS. The results (mean ± SD of 3 determinations) are) are expressed as percent input (left). Chromatin was analyzed for accessibility by nuclease digestion. The results (mean ± SD of 3 determinations) are expressed as % untreated chromatin (middle). NOTCH1 mRNA levels were analyzed by qRT-PCR (right). The results (mean ± SD of 3 determinations) are expressed as relative mRNA levels compared to that obtained for vehicle-treated cells (assigned a value of 1). **d** Soluble chromatin from BT-549/CshRNA and BT-549/JUNshRNA cells was precipitated with antibodies against the indicated proteins. The DNA samples were amplified by qPCR with primers for the *NOTCH1* pELS. The results (mean ± SD of 3 determinations) are expressed as percent input. **e** BT-549/tet-WDR5shRNA cells treated with vehicle or DOX were assessed for cell proliferation using the Alamar Blue Assay. The results (mean ± SD of 3 determinations) are expressed as percent reduction of Alamar Blue staining. **f** BT-549/tet-WDR5shRNA cells treated with vehicle or DOX were assayed for tumorsphere formation at 10 days. Scale bar: 100 μm. The results (mean + SD of 3 biological replicates) are expressed as the number of tumorspheres.

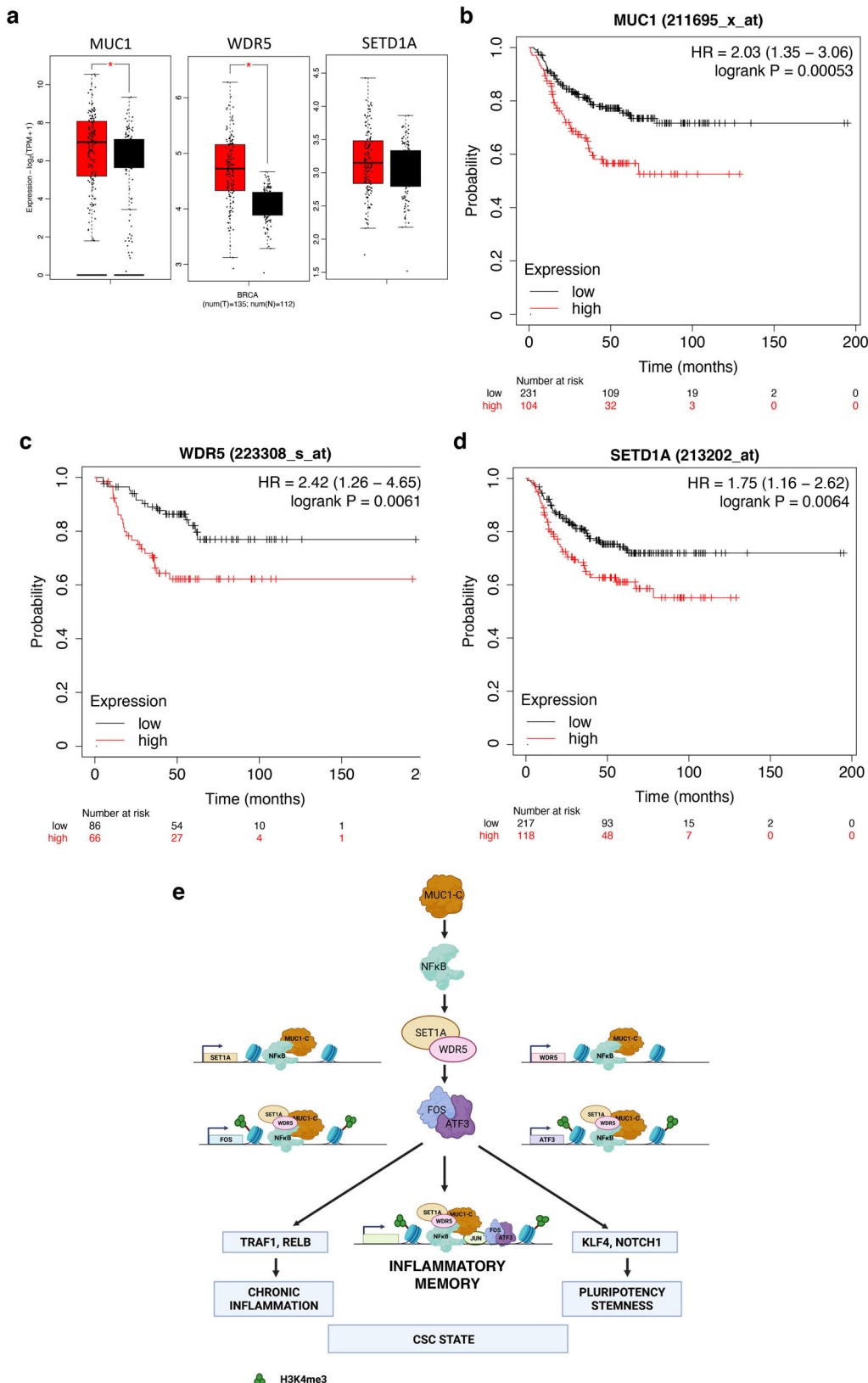

COMPASS. In the present work, we found that, unlike BAF/PBAF subunits, MUC1-C→E2F1 signaling has little, if any, role in activating genes encoding the SET1A/B COMPASS proteins. By contrast, we found that MUC1-C induces *SET1A*, and not *SET1B*, by an NF-κB p65-dependent pathway. MUC1-C binds directly to NF-κB p65 and promotes NF-κB-induced regulation of its target genes, which include effectors of inflammatory responses and the PRC2 EZH2 methyltransferase[12–14,16]. MUC1-C and NF-κB were detectable on *SET1A* at a pELS and, consistent with the effects of MUC1-C on NF-κB-mediated gene transactivation, MUC1-C silencing decreased NF-κB occupancy. SET1A/B HMT activity is dependent on the WRAD proteins[3]. Our results

**Fig. 8 MUC1-C→SET1A/WDR5 COMPASS signaling in TNBC tumors associates with poor clinical outcomes and in TNBC cells confers cancer progression. a** GEPIA2 analysis of MUC1, WDR5 and SET1A expression in 135 TNBC tumors from the TCGA-BRCA RNA-seq dataset as compared to normal breast tissue from TCGA and GTEx. The *P* value was set to 0.05. **b–d** Relapse-free survival in patients with TNBC tumors expressing high vs low levels of MUC1 (**b**), WDR5 (**c**) and SET1A (**d**). **e** MUC1-C binds directly to NF-κB p65 and contributes to the activation of NF-κB p65 target genes, such as *EZH2* and *SUZ12*, that encode components of the PRC2 complex. The present work demonstrates that MUC1-C/NF-κB occupy the *SET1A* and *WDR5* genes at PLSs and are necessary for SET1A and WDR5 expression. WDR5 is indispensable for SET1A complex assembly and HMT activity. Our results show that (i) MUC1-C associates with SET1A and WDR5, and (ii) MUC1-C, NF-κB, SET1A and WDR5 occupy the AP-1 encoding *FOS* and *ATF3* genes in association with increases in H3K4me3 and their expression. Of interest in this regard, FOS and ATF3 are activated by inflammatory stress in a potential feedback pathway that contributes to the activation of NF-κB and this COMPASS-associated pathway. In support of this notion, MUC1-C/NF-κB complexes associate with SET1A, WDR5 and AP-1 on target genes, such as *TRAF1* and *RELB*, in the TNF→NF-κB pathway that drives chronic inflammation in cancer cells. We also found that (i) MUC1-C/NF-κB associates with SET1A, WDR5 and AP-1 on *KLF4* and *NOTCH1*, and (ii) this complex is necessary for H3K4me3 deposition and their expression. These findings collectively support a model in which MUC1-C-induced activation of the SET1A/WDR5 COMPASS complex integrates inflammatory memory with the CSC state.

demonstrate that the MUC1-C→NF-κB pathway also activates *WDR5* and *RBBP5* (Fig. 8e). As found for *SET1A*, the *WDR5* gene contains a pELS that is occupied by MUC1-C/NF-κB complexes in a MUC1-C-dependent manner. These results supported a model in which MUC1-C and NF-κB integrate SET1A and WDR5, as well as RBBP5, expression in activating the SET1A HMT complex and inducing H3K4 methylation (Fig. 8e). SET1A and SET1B play redundant roles in implementing H3K4me1/3 deposition[28]. Therefore, the observations that MUC1-C drives SET1A and not SET1B expression indicated that MUC1-C might have little, if any, effect on bulk H3K4me1/3 levels, which we found were suppressed by MUC1-C silencing. Nonetheless, binding of SET1A and SET1B to the WDR5 WIN site is necessary for their HMT activities[3,6]. As a result and given that MUC1-C is necessary for WDR5 expression, targeting MUC1-C would be expected to suppress both SET1A/B HMT activity and thereby bulk H3K4me1/3 levels.

SET1A/B complexes are responsible for genome-wide H3K4 methylation in fly and mammalian cells[3]. MUC1-C first appeared in mammals to promote inflammatory, proliferative and remodeling responses that are associated with wound repair of barrier tissues[12,13]. Accordingly, MUC1-C-induced activation of the SET1A complex could conceivably represent an evolutionary adaptation for protecting barrier epithelial cells from loss of homeostasis that has been co-opted by cancer cells[12,13]. To define MUC1-C-regulated genes that are also controlled by the SET1A complex, we silenced WDR5, rather than SET1A, to avoid the potential redundancy of SET1A and SET1B activity. Comparison of MUC1-C and WDR5 gene signatures identified convergence in the HALLMARK TNFA SIGNALING VIA NFKB and HALLMARK INFLAMMATORY RESPONSE pathways. Among these MUC1-C- and WDR5-driven genes, we identified *TRAF1* and *RELB* and those encoding downstream immunosuppressive effectors, such as IL-6 and PTGS2, linking COMPASS to the regulation of chronic inflammation (Fig. 8e). We also identified the previously unrecognized MUC1-C- and WDR5-induced activation of AP-1 family members (Fig. 8e), which play pleotropic roles in inflammation, proliferation, and wound repair[35,36]. Notably, in this regard, NF-κB and AP-1 form transcriptional complexes that occupy contiguous NF-κB and AP-1 binding sites in promoting an inflammatory network linked to cancer progression[37]. We focused on FOS and ATF3 in that both are activated by inflammation, heterodimerize with JUN and, like MUC1-C, regulate the NF-κB transactivation function[37,39–43]. We confirmed that MUC1-C and WDR5 are necessary for the deposition of H3K4me3 on *FOS and ATF3* and for their expression (Fig. 8e). Subsequent studies will be needed to determine if MUC1-C and WDR5 regulate FOSL2, JDP2, JUNB and JUND by similar mechanisms.

MUC1-C induces the Yamanaka OSKM pluripotency factors in chronic inflammation and cancer cells[12,13]. In the present work, we found that WDR5 is necessary for increased KLF4 expression, which regulates pluripotency-associated enhancer networks characteristic of CSCs[58] (Fig. 8e). We also identified the previously unrecognized involvement of MUC1-C and WDR5 in the induction of SALL4, an effector of ESC lineage plasticity and stemness[53]. In addition, MUC1-C and WDR5 increased expression of the BMI1, LGR5 and NOTCH1 stemness factors. This association of MUC1-C and COMPASS with stemness was extended by the finding that MUC1-C is necessary for SET1A, WDR5 and RBBP5 expression in CSCs. MUC1-C activates (i) the BMI1/PRC1 complex in cancer cells[55], (ii) the LGR5 stemness-associated factor in the progression of colitis to colorectal cancer[51]; and (iii) NOTCH1 in TNBC, CRPC and other CSCs[18,20,57,62]; however, to our knowledge, little is known about the involvement of COMPASS in the regulation of these genes. Regarding *NOTCH1* activation, MUC1-C forms a complex with ARID1A/BAF on a pELS that is associated with increases in chromatin accessibility and NOTCH1 expression[20]. We found here that SET1A and WDR5 are also detectable on the *NOTCH1* pELS. By extension, silencing MUC1-C and WDR5 decreased the H3K4me3 mark, chromatin accessibility and NOTCH1 expression. JUN, FOS and ATF3 were also detectable on the *NOTCH1* pELS, which includes an AP-1 binding motif[20] (Fig. 8e). Unlike FOS and ATF3, MUC1-C had no apparent effect on JUN expression, but binds directly to the JUN transactivation domain[20]. JUN forms heterodimers with FOS and ATF3 and recruits BAF to AP-1 sites[63,64]. The present results demonstrate that JUN is also necessary for the recruitment of SET1A/WDR5 to the *NOTCH1* pELS (Fig. 8e). In concert with these results and the demonstration that silencing WDR5 decreases NOTCH1 expression, we found that, like MUC1-C, WDR5 is necessary for the self-renewal of cancer cells. These findings indicate that MUC1-C activates an NF-κB-driven pathway that regulates the SET1A/WDR5 COMPASS complex and integrates the activation of genes that promote intrinsic chronic inflammation, pluripotency and stemness, rather than that influenced by the tumor microenvironment.

Finally, increasing evidence indicates that MUC1-C is pan-cancer oncogenic protein[12,13,62]. MUC1-C has been linked to remodeling phases of wound repair that necessitate the capacity for lineage plasticity and self-renewal[12,13]. These responses are, in principle, reversible with repair. However, in settings of chronic inflammation, MUC1-C-driven responses can become established and promote the CSC state[12,13,62]. The present results support this notion by demonstrating that MUC1-C integrates chronic inflammation with epigenetic reprogramming by the COMPASS complex.

## Materials and methods

**Cell culture.** Human BT-549 TNBC cells (ATCC) were cultured in RPMI1640 medium (Thermo Fisher Scientific, Waltham, MA, USA) containing 10% fetal bovine serum (FBS; GEMINI Bio-Products, West Sacramento, CA, USA), 100 μg/ml streptomycin, 100 U/ml penicillin and 10 μg/ml insulin. Human MDA-MB-468 TNBC cells (ATCC) were cultured in Leibovitz's L-15 medium (Thermo Fisher Scientific) containing 10% FBS. Human SUM149 *BRCA1* mutant TNBC cells (ATCC) were grown in Ham's F-12 medium (Corning, Manassas, VA, USA) supplemented with 10 mM HEPES, 5% FBS, 100 μg/ml streptomycin, 100 U/ml penicillin, 5 μg/ml insulin and 1 μg/ml hydrocortisone. Human LNCaP-AI CRPC cells[29] were cultured in phenol red-free RPMI1640 medium (Thermo Fisher Scientific) containing 10% charcoal-stripped FBS (Millipore Sigma, Burlington, MA, USA). DU-145 CRPC cells (ATCC) were cultured in RPMI1640 medium (Corning Life Sciences, Corning, NY, USA) containing 10% heat-inactivated FBS. Cells were treated with the NF-κB inhibitor BAY11-7082 (S2913; Selleckchem, Houston, TX, USA) and MUC1-C inhibitor GO-203[12]. Cells were maintained in culture for 3–4 months. Authentication of the cells was performed by short tandem repeat (STR) analysis. Cells were monitored for mycoplasma contamination using the MycoAlert Mycoplasma Detection Kit (Lonza, Rockland, ME, USA).

**Gene silencing and rescue.** MUC1shRNA (MISSION shRNA TRCN0000122938), WDR5shRNA (MISSION shRNA TRCN0000118047) and a control scrambled shRNA (CshRNA) (Millipore Sigma) were inserted into pLKO-tet-puro (Plasmid #21915; Addgene, Cambridge, MA, USA). The CshRNA, MUC1shRNA, MUC1shRNA#2 (MISSION shRNA TRCN0000430218) and NF-κBshRNA (MISSION shRNA TRCN0000014687) were produced in HEK293T cells as described[29]. MUC1-C cDNA and Flag-tagged MUC1-CD[65] were inserted into pInducer20 (Plasmid #44012, Addgene)[66]. Cells transduced with the vectors were selected for growth in 1–2 μg/ml puromycin or 100 μg/ml geneticin. For inducible gene silencing, cells were treated with 0.1% DMSO as the vehicle control or 500 ng/ml doxycycline (DOX; Millipore Sigma).

**Immunoblot analysis.** Whole-cell lysates were prepared in RIPA buffer containing protease inhibitor cocktail (Thermo Fisher Scientific). Immunoblotting was performed with anti-MUC1-N (#14161, Cell Signaling Technology (CST), Danvers, MA, USA), anti-MUC1-C (#16564S, 1:1000 dilution, CST), anti-H3K4me1 (#ab8895, 1:1000 dilution, Abcam), anti-H3K4me3 (#ab8580, 1:1000 dilution; Abcam), anti-H3 (#9715S, 1:1000 dilution; CST) anti-SET1A (#61702S, 1:1000 dilution; CST), anti-SET1B (#44922S, 1:1000 dilution; CST), anti-WDR5 (#13105S, 1:1000 dilution; CST), anti-RBBP5 (#13171S, 1:1000 dilution; CST), anti-ASH2L (#ABE1972, 1:1000 dilution; Millipore), anti-NF-κB p65 (#8242S, 1:1000 dilution; CST), anti-ATF3 (33593S, 1:1000 dilution; CST), anti-c-FOS (#2250S, 1:1000 dilution; CST), anti-c-JUN (#3742S, 1:1000 dilution; CST), anti-JUNB (#3753S, 1:1000 dilution; CST), anti-JUND (#5000S, 1:1000 dilution; CST), anti-JDP2 (#ab40916, 1:1000 dilution; Abcam), anti-Tubulin (#2144S, 1:1000 dilution; CST), anti-GAPDH (#2118, 1:1000 dilution; CST) and anti-ß-actin (A5441; 1:50000 dilution; Sigma, St. Louis, MO, USA).

**Quantitative reverse-transcription PCR (qRT-PCR).** Total cellular RNA was isolated using Trizol reagent (Thermo Fisher Scientific). cDNAs were synthesized using the High-Capacity cDNA Reverse Transcription Kit (Applied Biosystems, Grand Island, NY, USA). The cDNA samples were amplified as described[67]. Primers used for qRT-PCR are listed in Supplementary Table S1.

**Click-iT Nascent RNA Assay.** Nascent RNA labeling with ethylene uridine (EU) was performed using the Click-iT Nascent RNA Capture kit (#C10365; Invitrogen, Waltham, MA, USA) according to the manufacturer's protocol. Briefly, cells were pulsed with 0.5 mM EU for 24 h. Nascent transcripts were captured from isolated total RNA on streptavidin magnetic beads. cDNA synthesis was performed on the beads using the High-Capacity cDNA Reverse Transcription Kit followed by qRT-PCR analysis.

**Coimmunoprecipitation of nuclear proteins.** Nuclear lysates were isolated as described[18]. DNA was digested by incubation in 20 U/ml DNase for 30 min at 37 °C. Nuclear proteins were incubated with anti-MUC1-C (#MA5-11202, Thermo Fisher Scientific) at 4 °C overnight and then precipitated with Dynabeads Protein G (10003D; Thermo Fisher Scientific) for 2 h at 4 °C. Beads were washed and then suspended in a sample loading buffer as described[18].

**Chromatin immunoprecipitation (ChIP).** ChIP was performed on cells crosslinked with 1% formaldehyde for 5 min at 37 °C, quenched with 2 M glycine, washed with PBS, and then sonicated in a Covaris E220 sonicator to generate 300–600 bp DNA fragments. Immunoprecipitation was performed using a control IgG (3900S, CST) and antibodies against MUC1-C (#16564S, CST), SET1A (#61702S, CST), WDR5 (#13105S, CST), NF-κB p65 (#ab16502, Abcam), H3K4me3 (#ab8580, Abcam), JUN (#ab32137, Abcam), c-FOS (#2250S, CST) and ATF3 (#33593S, CST). Precipitated DNAs were detected by PCR using primers listed in Supplementary Table S2. Quantitation was performed on immunoprecipitated DNA using SYBR-green and the CFX384 real-time PCR machine (Bio-Rad, USA). Data are reported as percentage of input DNA for each sample.

**RNA-seq analysis.** Total RNA from cells cultured in triplicates was isolated using Trizol reagent (Invitrogen) as described[21,23,29]. TruSeq Stranded mRNA (Illumina, San Diego, CA, USA) was used for library preparation. Raw sequencing reads were aligned to the human genome (GRCh38.74) using STAR. Raw feature counts were normalized and differential expression analysis using DESeq2 as described[21,23,29]. Differential expression rank order for subsequent GSEA was performed using the fgsea (v1.8.0) package in R. Hallmark Gene Sets were queried through the Molecular Signatures Database.

**ATAC-seq.** ATAC-seq libraries were generated from three biologically independent replicates per condition. Library preparation and quality control were performed as described[20,68]. The raw ATAC-seq data were processed using the pipeline (https://github.com/macs3-project/genomics-analysis-pipelines). To generate the signal tracks for the Integrative Genome Browser snapshots, we used MACS2 to pileup the aligned ATAC-seq read pairs and normalized the pileup values by the million read depth of each library as described[20].

**Chromatin accessibility assay.** DNAse1 chromatin accessibility assays were performed on chromatin isolated as described[20]. Aliquots of chromatin were left untreated or digested with 3 U/100 μl DNase I (Promega, Madison, WI, USA) for 5 min at room temperature as described[20]. DNA was purified and amplified by qPCR using primers listed in Supplementary Table S2. qPCR results were analyzed according to the formula $100/2^{Ct \, (DNase \, I) \, - \, Ct \, (no \, DNase \, I)}$.

The data were normalized to input DNA without DNase I treatment.

**Cell proliferation assays.** Cells (6000) were seeded per well in 96 well plates. Cell proliferation was assessed using the Alamar Blue Assay (Thermo Scientific, Rockford, IL, USA). Fluorescence intensity (560 nm excitation/590 nm emission) was measured in quintuplicate.

**Tumorsphere formation assays.** Single-cell suspensions were cultured in MammoCult Human MediumKit (Stemcell Technologies) at a density of 5000 cells per well of a 6-well ultralow attachment culture plate (Corning) for 10 days as described[20]. Tumorspheres with a diameter >50 microns were counted under an inverted microscope in triplicate wells.

**Analysis of gene expression in TNBC tumors and normal breast tissue.** GEPIA2 analysis of gene expression was performed using TCGA-BRCA TNBC RNA-seq data and TCGA/GTEx data from normal breast tissue[61].

**Survival analysis of patients with TNBC tumors.** Survival curves based on MUC1, WDR5 and SETD1A expression levels were generated using the Kaplan–Meier Plotter (http://kmplot.com/analysis/). Breast cancer patients sorted by ER-negative, HER2-negative, PR-negative and basal phenotype (PAM50) and treated with chemotherapy were included in this analysis. The statistical difference was calculated using the log-rank test. A Cox proportional hazards regression model was used to assess the prognostic value of gene expression levels.

**Statistics and reproducibility.** Each experiment was performed with at least three independent biologic replicates. Data are expressed as the mean ± SD. The unpaired Student's $t$-test was used to examine differences between the two groups. A $P$ value of <0.05 was considered a statistically significant difference. GraphPad Prism 8 was used for all statistical analyses. Asterisks represent *$P \leq 0.05$, **$P \leq 0.01$, ***$P \leq 0.001$, ****$P \leq 0.0001$ with CI = 95%.

### Data availability

The RNA-seq data have been deposited in the GEO database under accession codes GSE164141 and GSE203055[23–25]. The ATAC-seq data have been deposited in the GEO database under accession code GSE180599[20]. The original immunoblots presented in this study are available in Supplementary Fig. S9. Source data can be found in Supplementary Data 1. All other data are available from the corresponding author upon reasonable request.

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

## Acknowledgements

Research reported in this publication was supported by the National Cancer Institute of the National Institutes of Health under grant numbers CA232979 and OD024973 awarded to S.L. and CA97098, CA233084 and CA262991 awarded to D.K.

## Author contributions

Conceptualization: A.B., D.K. Methodology: A.B., T.L. Investigation: A.B., K.W., N.Y., Y.M., S.I., T.D. Bioinformatics analysis: A.B., A.F., T.L., M.D.L., S.L. Writing—original draft: D.K. Writing—review and editing: A.B., M.D.L., D.K. Funding acquisition: S.L., D.K.

## Competing interests

D.K. has equity interests in Genus Oncology and Hillstream Biopharma and is a paid consultant to Reata and CanBas. The other authors declare no competing interests.
