## [Peer Review File · Communications Biology]

Reviewers' comments:

Reviewer #1 (Remarks to the Author):

The current manuscript entitled 'MUC1-C INTERSECTS CHRONIC INFLAMMATION WITH EPIGENETIC REPROGRAMMING BY REGULATING THE SET1A COMPASS COMPLEX IN CANCER PROGRESSION' claims that MUC1-C regulates inflammation and pluripotency in cancer cells via the NF- κ B and AP-1 pathways. There are several approaches and interesting results. Please see below my comments based on the results.

Fig1A: Silencing of MUC1C causes reduction of H3K4me3 over time, but the levels of H3K4me1 are reduced until Day 7 and then they come back. How can this be explained?

Fig1B+1C:

- Why the incubation for the experiments was for 7 days and not a relevant time course
- The calculations for the mRNA levels use as reference the Dox- sample at the same time point (Day 7). We don't know though the mRNA levels of these proteins at time 0. The calculations need to be done in another way so we can see how the mRNA levels are affected over time (7-day period) and have separate graphs/plots for the 2 conditions (with and without doxycycline)
- Similarly, western blots of time 0 and Day7 should be presented at the 2 conditions (with and without doxycycline)

Fig1E: there are 2 bands in the H3 western. How this can be explained?

Fig2E: - Treatment with the GO-203 inhibitor. It would be necessary to see the mRNA and protein levels of the MUC1-C.

Fig2G:

- It is not clear why in this experiment the tumorsphere cells were analysed after 7 days of DOX treatment. Is there an experiment that shows that the 2D and 3D cultures respond in a similar way during DOX treatment?

Fig3: NF- κ B is a known dimeric transcription factor. IN the canonical pathway (that Bay 11 inhibits) p65 forms heterodimers with p50, cRel, even RelB. It also forms homodimers. There is no indication throughout the manuscript about the levels, fate, and function of the possible p65 partners.

Fig3 legend: D: Schema correction to Scheme

Fig4: Similarly, to Fig3 comment, since many members of the AP-1 family (FOS, JUNB, JUND, JDP2, ATF3 and BATF3) are affected by MUC1 silencing, it is not checked what happens to the protein levels of the other family members and which dimers are inhibited in the ChIp experiment (Fig4F and 4G). IT is not recommended to make statements about the function of a transcription factor dimer by testing only one of the partners.

Fig6C and D : The Fold change of the 4 target genes is not similar when Muc1 is silenced versus WDR5 silencing. Eg SALL4 is much more affected when MUC1 is silenced (Fig6C). I would expect similar reduction. Is there explanation for this discrepancy?

The bioinformatics analysis (GSEA results) identified Inflammatory pathways related to the NF- κ B and TNF signalling (Fig5C). Nevertheless, it is not clear how 'Inflammation' is linked to the specific cell lines tested in the current study and the culture conditions. Based on the M&M, the cells were cultured in standard conditions.

Reviewer #2 (Remarks to the Author):

Authors:

Drs. Atrayee Bhattacharya, Atsushi Fushimi, Nami Yamashita, Yoshihiro Morimoto, Satoshi Ishikawa, Tatsuaki Daimon, Tao Liu, Song Liu, Mark D. Long, and Donald Kufe,

Title:

MUC1-C intersects chronic inflammation with epigenetic reprogramming by regulating the SET1a Compass complex in cancer progression.

Outline:

Effects of expression or suppression of MUC1-C on the levels of a polycomb gene family member, one of epigenetic regulatory molecules, were examined and importance of MUC1-C for the expression of SET1A associated with Histone methylation was shown. By searching the events upstream and downstream from SET1A, importance of NF- κ B and AP-1 pathway was revealed. The authors claim that these mechanisms play a role in the stemness and inflammatory characteristics of triple negative breast cancer cells and other malignant cells.

Critique:

This paper presents a very interesting possibility that MUC1-C has an epigenetic impact. Although considerable amount of work was invested, the conclusions as indicated in the title of the paper and in the last sentence of the Discussion do not seem to be supported by the experiments.

Specific points to improve the manuscript:

- (1) In this report, MUC1-C is shown by the western blotting after separated from its extracellular domain. It is not clear whether MUC1-C separated from the extracellular domain is present in the cells or generated during the detection process. This point, i.e. possible association with the extracellular domain even in the cells, should be clarified by additional experiments.
- (2) It is almost impossible to follow the logical consequences due to heavy uses of abbreviated terms. The paper should be rewritten to be understandable to a wide range of readers.
- (3) The experiment revealed that a signal axis of MUC1 > NF- κ B > WDR5/RBBP5 > SET1A > AP1 is present in TNBC cells and some other carcinoma cells. There are many biological questions which should be answered before claiming that the findings are important from the cancer biology context. One of the most important points to be included in the revised paper would be how this process is related to anything unique to the biology of TNBC? Although the authors may want to claim that the process plays important role in the progression of many different types of cancer and TNBC is just a model, it is necessary to prove and present that this axis is actually important in the TNBC progression. Proofs at the cell levels or pathological specimen levels should be added.
- (4) The authors' claim that MUC1-C promotes pluripotency sounds somewhat odd. If it promotes generation of cancer cell heterogeneity, it should not drive unidirectional progression toward increased malignancy unless the cells are under influences of host environment. This point should be covered in the Discussion
- (5) The authors claim that the process induces chronic inflammation. However, inflammation is a process caused by host responses. Inflammation cannot be argued without considering associated immune and inflammatory cells in cancer tissues. Therefore, the current descriptions are not convincing. Consultations by immunologists are highly recommended.
- (6) Survival differences in TNBC patients according to the expression of MUC1, WDR5, and SET1A are shown to emphasize the clinical relevance of the work. However, a general consensus is that MUC1 mRNA levels and MUC1 glycoprotein levels do not correspond in tumor specimens. It is obvious that, MUC1 mRNA and MUC1-C protein levels are not expected to correspond to each other. It would be nice to obtain actual data on the levels of these three molecules using the same cohort at the protein level, if the authors would like to include this part in the Results.
- (7) MUC1 is a molecule highly expressed rather ubiquitously among a variety of epithelia. Is the process presented here operating in these somatic cells? Is the generation of MUC1-C limited to some types of malignant cells? Answers to these fundamental questions should be discussed.

Conclusions:

The paper is acceptable with major revisions after additional experiments to directly address the authors' claims. Also, extensive rewriting is highly recommended.

Reviewer #3 (Remarks to the Author):

Previous work by the Kufe lab has established links of the oncogene MUC1-C to epigenetic and transcriptional reprogramming by modulating DNA methylation, histone modifications and chromatin remodeling in cancer cells. A common key mechanism seems that overexpression of MYC1-C (as seen in a majority of cancers) acts at chromatin to alter the expression of key transcription factors (e.g. NF- κ B) and selective subunits of the corresponding cofactor complexes (e.g. DNMTs, PRCs, NuRD, SWI/SNF).

In the current manuscript, the authors identify a hitherto unknown NF- κ B - MUC1-C - SET1A/WDR5/COMPASS - H3K4me3 - AP-1 - KLF4/Notch1 pathway that links chronic inflammation to epigenetic reprogramming in triple negative breast cancer cells. The techniques used are appropriate and include amongst others the analysis of gene expression (RNA-seq, qPCR), protein levels (western blot) and interactions (Co-IP), chromatin accessibility (ATAC-seq, DNase footprints), chromatin occupancy by TFs and cofactors (ChIP-qPCR), all upon shRNA-mediated depletion of MUC1-C and additional proteins. Most of the experiments are well designed and controlled, although the flow of the presentation and the selection of the data is sometimes difficult to follow.

The following few comments, suggestions and questions should be addressed to improve the study towards publication.

1. Given that the oncogene MUC1-C is overexpressed in many cancers, what is the rationale in depleting it in TNBC cells? Wouldn't overexpression experiments and comparisons of existing breast cancer cells lines with low vs. high MUC1-C expression more relevant models?
2. It seems that the current study utilizes RNA-seq data and ATAC-seq data generated in their previous studies (PMID: 33495298; PMID: 35681561), comparing differentially expressed genes and differentially accessible chromatin regions (e.g. promoters, enhancers) upon MUC1-C depletion in in triple-negative breast cancer (TNBC) cell lines. This should be clearly mentioned in the methods and results sections, with reference to these papers. While the GEO accession numbers for the RNA-seq data are provided, I miss the ATAC-seq data (triplicates, as stated). These should be made available to the readers with acceptance.
3. In Fig. 1, instead of bulk western blots for H3K4 modifications upon MUC1-C depletion, the authors should perform ChIP-seq for H3K4me1 (enhancer mark) and H3K4me3 (active promoter mark) to establish functional links to transcription and gene expression (which they analyze by RNA-seq).
4. The intersection of ChIP-seq, ATAC-seq and RNA-seq data would substantially improve the genome-wide analysis and interpretation of MUC1-C functions, given the multiple target genes and factors already identified in previous studies. The comment is relevant for all Figures, analyzing currently only selected genes (RT-qPCR, ATAC-seq genome browser shots) or only the RNA-seq data (Fig. 4). It should be easily possible to perform a comparative genome-wide analysis of RNA-seq and ATAC-seq data. This would address important mechanistic and clinical relevant questions, such as what is the relationship between chromatin accessibility changes and gene expression changes upon MUC1-C depletion, and which promoters/enhancers are direct targets of MUC1-C? If there are discrepancies, they are also of interest, because a change of accessibility without corresponding transcriptional changes may hint at epigenetic memory mechanisms.
5. What is the gene/locus specificity of MUC1-C? Is there published ChIP-seq data (or have the

authors tried to generate some) for MUC1-C in TNBC cells which the authors could overlay with SET1A/WDR5/COMPASS? For the moment, it is unclear why expression of some COMPASS subunits (SET1A, WDR5, RBBP5) are regulated by MUC1-C, while others are not (SET1B, ASH2L). If it is simply the presence of p65/NF- κ B binding sites, the hypothesis should be supported by analysis of published p65 ChIP-seq data at the loci of these genes. In Fig. 3 (p65 depletion), SET1B should be included along with SET1A. Additionally, along with comments 3 and 4, please show that ATAC-seq genome browser shot for SET1A (transcription downregulated upon MUC1-C depletion) vs. SET1B (transcription not regulated upon MUC1-C depletion). The authors could also analyze by ChIP-qPCR whether MUC1-C occupies SET1A vs. SET1B promoter and enhancers, revealing perhaps how gene selectivity could be achieved.

6. Western blots show that depletion of MUC1-C reduces expression of SET1A, WDR5, RBBP5, several AP-1 members, but not of SET1B, ASH2L, Jun. Is this also seen in the RNA-seq data, are they part of the 100 downregulated genes in Fig. 4B/C? If yes, they could be highlighted in the Figure, if not, it should be discussed.

7. The introduction should be improved to concisely summarize what is already known about the genomic actions of MUC1-C, and what are the rationale, key questions, hypotheses of the current study? A meaningful rationale would be, for example, to start with the RNA-seq analysis to identify COMPASS subunits (SET1A, WDR5) as hitherto unknown MUC1-C targets (control vs. depletion, high vs. low expression). A weaker rationale is the possible link between MUC1-C and H3K4me3: H3K4me3 is an established promoter (sometimes also enhancer) activity mark, thus the altered expression of any transcription factor and cofactor that changes gene transcription will cause changes in H3K4me3 at promoters (enhancers) of regulated genes. It could be correlation, cause or consequence, and would in most cases not require any direct link of the (co-)factor to COMPASS.

Responses to the Reviewers

Reviewer #1:

The current manuscript entitled 'MUC1-C INTERSECTS CHRONIC INFLAMMATION WITH EPIGENETIC REPROGRAMMING BY REGULATING THE SET1A COMPASS COMPLEX IN CANCER PROGRESSION' claims that MUC1-C regulates inflammation and pluripotency in cancer cells via the NF- κ B and AP-1 pathways. There are several approaches and interesting results. Please see below my comments based on the results.

(1) Fig1A: Silencing of MUC1C causes reduction of H3K4me3 over time, but the levels of H3K4me1 are reduced until Day 7 and then they come back. How can this be explained?

In response, we have added the following description to the Results section "Analysis of BT-549 TNBC cells demonstrated that silencing MUC1-C for 7 days is associated with decreases in bulk H3K4me1 and H3K4me3 levels (Fig. 1A). These decreases in H3K4me3 were sustained over 21 days; whereas, H3K4me1 levels were upregulated on days 14 and 21, which were attributed to a potential feedback mechanism (Fig. 1A)".

(2) Fig1B+1C:

- Why the incubation for the experiments was for 7 days and not a relevant time course
- The calculations for the mRNA levels use as reference the Dox-sample at the same time point (Day 7). We don't know though the mRNA levels of these proteins at time 0. The calculations need to be done in another way so we can see how the mRNA levels are affected over time (7-day period) and have separate graphs/plots for the 2 conditions (with and without doxycycline)
- Similarly, western blots of time 0 and Day7 should be presented at the 2 conditions (with and without doxycycline)

As requested, we have included a 7 day time course for regulation of MUC1-C, SET1A and SET1B mRNA (new Supplemental Fig. S1C) and protein (new Supplemental Fig. S1D) levels in the absence and presence of DOX.

(3) Fig1E: there are 2 bands in the H3 western. How this can be explained?

The two bands in the H3 IB are the result of degradation. Accordingly, we have repeated the IB with fresh lysate, which shows a single band as observed in our other IBs (revised Fig. 1E).

(4) Fig2E: - Treatment with the GO-203 inhibitor. It would be necessary to see the mRNA and protein levels of the MUC1-C.

The effects of GO-203 on MUC1-C mRNA and protein levels were included in Fig. 1F.

(5) Fig2G: It is not clear why in this experiment the tumorsphere cells were analysed after 7 days of DOX treatment. Is there an experiment that shows that the 2D and 3D cultures respond in a similar way during DOX treatment?

In response, we have included the statement "Using the same conditions for BT-549 2D cells (Figs. 1B and 2A), silencing MUC1-C for 7 days in the BT-549 3D mammosphere cells had similar effects on the regulation of MUC1-C, SET1A, WDR5 and RBBP5 expression (Fig. 2G), supporting a role for MUC1-C in regulating COMPASS in CSCs."

(6) Fig3: NF- κ B is a known dimeric transcription factor. IN the canonical pathway (that Bay 11 inhibits) p65 forms heterodimers with p50, cRel, even RelB. It also forms homodimers. There is no indication throughout the manuscript about the levels, fate, and function of the possible p65 partners.

MUC1-C binds directly to NF- κ B p65 (RELA) and regulates the NF- κ B p65 transactivation function; whereas little is known about involvement with NF- κ B p50, c-REL or RELB. Accordingly, we focused on NF- κ B p65 and have clarified this point in the Results section.

(7) Fig3 legend: D: Schema correction to Scheme

As suggested, we have corrected "Schema" to "Scheme".

(8) Fig4: Similarly, to Fig3 comment, since many members of the AP-1 family (FOS, JUNB, JUND, JDP2, ATF3 and BATF3) are affected by MUC1 silencing, it is not checked what happens to the protein levels of the other family members and which dimers are inhibited in the ChIP experiment (Fig4F and 4G). IT is not recommended to make statements about the function of a transcription factor dimer by testing only one of the partners.

As requested and in addition to FOS, we have included analysis of MUC1-C- and WDR5-dependent regulation of JUNB, JUND and JDP2 (Supplemental Fig. S4I), which indicated that the effects of the MUC1-C/WDR5 pathway are more pronounced for FOS (Fig. 4E). We therefore focused on the FOS gene as stated in the text.

In regard to the ChIP studies, we investigated occupancy of the FOS pELS region by MUC1-C, NF- κ B, SET1A and WDR5 (Figs. 4F and 4G). Of note, these ChIP studies were not designed to assess occupancy of AP-1 family members.

(9) Fig6C and D: The Fold change of the 4 target genes is not similar when Mucl is silenced versus WDR5 silencing. Eg SALL4 is much more affected when MUC1 is silenced (Fig6C). I would expect similar reduction. Is there explanation for this discrepancy?

Yes, MUC1-C may be contributing to the regulation of factors other than WDR5 in activating SALL4 expression. A statement to this effect has been included in the Results section.

(10) The bioinformatics analysis (GSEA results) identified Inflammatory pathways related to the NF- κ B and TNF signalling (Fig5C). Nevertheless, it is not clear how 'Inflammation' is linked to the specific cell lines tested in the current study and the culture conditions. Based on the M&M, the cells were cultured in standard conditions.

In response, MUC1-C contributes to intrinsic activation of chronic inflammation in TNBC cells, precluding the need for stimulation with pro-inflammatory cytokines.

Reviewer #2:

Outline:

Effects of expression or suppression of MUC1-C on the levels of a polycomb gene family member, one of epigenetic regulatory molecules,

were examined and importance of MUC1-C for the expression of SET1A associated with Histone methylation was shown. By searching the events upstream and downstream from SET1A, importance of NF- κ B and AP-1 pathway was revealed. The authors claim that these mechanisms play a role in the stemness and inflammatory characteristics of triple negative breast cancer cells and other malignant cells.

Critique:

This paper presents a very interesting possibility that MUC1-C has an epigenetic impact. Although considerable amount of work was invested, the conclusions as indicated in the title of the paper and in the last sentence of the Discussion do not seem to be supported by the experiments.

Specific points to improve the manuscript:

(1) In this report, MUC1-C is shown by the western blotting after separated from its extracellular domain. It is not clear whether MUC1-C separated from the extracellular domain is present in the cells or generated during the detection process. This point, i.e. possible association with the extracellular domain even in the cells, should be clarified by additional experiments.

In response, we demonstrate that MUC1-C, but not MUC1-N, is detectable in the nucleus (new Supplemental Fig. S1A), indicating that MUC1-N is not directly involved in the regulation of COMPASS and epigenetic reprogramming. A statement to this effect is included in the Results section.

(2) It is almost impossible to follow the logical consequences due to heavy uses of abbreviated terms. The paper should be rewritten to be understandable to a wide range of readers.

As requested, we have included full descriptions of abbreviated terms when they are first introduced in the text.

(3) The experiment revealed that a signal axis of MUC1 > NF- κ B > WDR5/RBBP5 > SET1A > AP1 is present in TNBC cells and some other carcinoma cells. There are many biological questions which should be answered before claiming that the findings are important from the cancer biology context. One of the most important points to be included in the revised paper would be how this process is related to anything unique to the biology of TNBC? Although the authors may want to claim that the process plays important role in the progression of

many different types of cancer and TNBC is just a model, it is necessary to prove and present that this axis is actually important in the TNBC progression. Proofs at the cell levels or pathological specimen levels should be added.

TNBCs are aggressive malignancies with relatively high levels of CSCs functionally characterized by the capacity for self-renewal, tumorigenicity and therapeutic resistance. MUC1-C has been implicated in driving the TNBC CSC state by mechanisms that remain incompletely understood. The present studies have focused on MUC1-C-driven epigenetic reprogramming by the COMPASS complex as a potential mechanism that contributes to TNBC progression. Our results demonstrate that MUC1-C regulates WDR5-mediated epigenetic reprogramming and that this pathway is necessary for expression of stemness-associated genes and for self-renewal of TNBC CSCs. These findings are of importance in understanding the basis for TNBC progression. Statements to this effect have been included in the Introduction.

(4) The authors' claim that MUC1-C promotes pluripotency sounds somewhat odd. If it promotes generation of cancer cell heterogeneity, it should not drive unidirectional progression toward increased malignancy unless the cells are under influences of host environment. This point should be covered in the Discussion.

As suggested, we have addressed this point in the Discussion by stating: "MUC1-C integrates activation of genes which promote intrinsic chronic inflammation, pluripotency and stemness, rather than that influenced by the tumor microenvironment".

(5) The authors claim that the process induces chronic inflammation. However, inflammation is a process caused by host responses. Inflammation cannot be argued without considering associated immune and inflammatory cells in cancer tissues. Therefore, the current descriptions are not convincing. Consultations by immunologists are highly recommended.

Previous work has demonstrated that MUC1-C drives TNBC cell intrinsic chronic inflammation by activation of the (i) type II IFN pathway, (ii) pattern recognition receptors and type I IFN pathway, and (iii) downstream IFN stimulated genes (ISGs) that promote DNA damage resistance and immune evasion. These findings, as highlighted in the Introduction, were uncovered in collaboration with the Roswell Park immunology team, who are members the Moonshot Immuno-Oncology Translational Network and are also coauthors on the present work.

(6) Survival differences in TNBC patients according to the expression of MUC1, WDR5, and SET1A are shown to emphasize the clinical relevance of the work. However, a general consensus is that MUC1 mRNA levels and MUC1 glycoprotein levels do not correspond in tumor specimens. It is obvious that, MUC1 mRNA and MUC1-C protein levels are not expected to correspond to each other. It would be nice to obtain actual data on the levels of these three molecules using the same cohort at the protein level, if the authors would like to include this part in the Results.

To our knowledge, TNBC proteomic and survival data are not available for these analyses.

(7) MUC1 is a molecule highly expressed rather ubiquitously among a variety of epithelia. Is the process presented here operating in these somatic cells? Is the generation of MUC1-C limited to some types of malignant cells? Answers to these fundamental questions should be discussed.

The evidence indicates that MUC1-C is pan-cancer oncogenic protein. MUC1-C has been linked to remodeling phases of wound repair that necessitate the capacity for lineage plasticity and self-renewal. These responses are, in principle, reversible with repair. However, in the setting of chronic inflammation, these responses can become established and promote cancer progression. The present results support this notion by demonstrating that MUC1-C integrates chronic inflammation with epigenetic reprogramming by the COMPASS complex. These points are highlighted in the Discussion.

Conclusions:

The paper is acceptable with major revisions after additional experiments to directly address the authors' claims. Also, extensive rewriting is highly recommended.

Reviewer #3

Previous work by the Kufe lab has established links of the oncogene MUC1-C to epigenetic and transcriptional reprogramming by modulating DNA methylation, histone modifications and chromatin remodeling in cancer cells. A common key mechanism seems that overexpression of MYC1-C (as seen in a majority of cancers) acts at chromatin to alter the expression of key transcription factors (e.g. NF- κ B) and selective subunits of the corresponding cofactor complexes (e.g. DNMTs, PRCs, NuRD, SWI/SNF).

In the current manuscript, the authors identify a hitherto unknown NF- κ B - MUC1-C - SET1A/WDR5/COMPASS - H3K4me3 - AP-1 - KLF4/Notch1 pathway that links chronic inflammation to epigenetic reprogramming in triple negative breast cancer cells. The techniques used are appropriate and include amongst others the analysis of gene expression (RNA-seq, qPCR), protein levels (western blot) and interactions (Co-IP), chromatin accessibility (ATAC-seq, DNase footprints), chromatin occupancy by TFs and cofactors (ChIP-qPCR), all upon shRNA-mediated depletion of MUC1-C and additional proteins. Most of the experiments are well designed and controlled, although the flow of the presentation and the selection of the data is sometimes difficult to follow.

The following few comments, suggestions and questions should be addressed to improve the study towards publication.

1. Given that the oncogene MUC1-C is overexpressed in many cancers, what is the rationale in depleting it in TNBC cells? Wouldn't overexpression experiments and comparisons of existing breast cancer cells lines with low vs. high MUC1-C expression more relevant models?

TNBCs are aggressive malignancies with relatively high levels of CSCs functionally characterized by the capacity for self-renewal, tumorigenicity and therapeutic resistance. MUC1-C has been implicated in driving the TNBC CSC state by mechanisms that remain incompletely understood. The present studies have focused on MUC1-C-driven epigenetic reprogramming by the COMPASS complex as a potential mechanism that contributes to TNBC progression. Our results demonstrate that MUC1-C regulates WDR5-mediated epigenetic reprogramming and that this pathway is necessary for expression of stemness-associated genes and for self-renewal of TNBC CSCs. These findings are of importance in understanding the basis for TNBC progression. Statements to this effect have been included in the Introduction.

2. It seems that the current study utilizes RNA-seq data and ATAC-seq data generated in their previous studies (PMID: 33495298; PMID: 35681561), comparing differentially expressed genes and differentially accessible chromatin regions (e.g. promoters, enhancers) upon MUC1-C depletion in in triple-negative breast cancer (TNBC) cell lines. This should be clearly mentioned in the methods and results sections, with reference to these papers. While the GEO accession numbers for the RNA-seq data are provided, I miss the ATAC-seq data (triplicates, as stated). These should be made available to the readers with acceptance.

As requested, we have clearly identified and referenced the previous RNA-seq and ATAC-seq studies in the Methods and Results sections. In addition, we have included the GEO accession number for the ATAC-seq data.

3. In Fig. 1, instead of bulk western blots for H3K4 modifications upon MUC1-C depletion, the authors should perform ChIP-seq for H3K4me1 (enhancer mark) and H3K4me3 (active promoter mark) to establish functional links to transcription and gene expression (which they analyze by RNA-seq).

We would respectfully contend that ChIP-seq studies are beyond the scope of the present work and, as stated in the Discussion, will be necessary for extending these findings.

4. The intersection of ChIP-seq, ATAC-seq and RNA-seq data would substantially improve the genome-wide analysis and interpretation of MUC1-C functions, given the multiple target genes and factors already identified in previous studies. The comment is relevant for all Figures, analyzing currently only selected genes (RT-qPCR, ATAC-seq genome browser shots) or only the RNA-seq data (Fig. 4). It should be easily possible to perform a comparative genome-wide analysis of RNA-seq and ATAC-seq data. This would address important mechanistic and clinical relevant questions, such as what is the relationship between chromatin accessibility changes and gene expression changes upon MUC1-C depletion, and which promoters/enhancers are direct targets of MUC1-C? If there are discrepancies, they are also of interest, because a change of accessibility without corresponding transcriptional changes may hint at epigenetic memory mechanisms.

In response, our previous work involved a comprehensive genome-wide analysis of the RNA-seq and ATAC-seq data, which demonstrated that MUC1-C-induced DARs align with genes regulated by the AP-1 family. These findings have been highlighted in the Results section.

5. What is the gene/locus specificity of MUC1-C? Is there published ChIP-seq data (or have the authors tried to generate some) for MUC1-C in TNBC cells which the authors could overlay with SET1A/WDR5/COMPASS? For the moment, it is unclear why expression of some COMPASS subunits (SET1A, WDR5, RBBP5) are regulated by MUC1-C, while others are not (SET1B, ASH2L). If it is simply the presence of p65/NF- κ B binding sites, the hypothesis should be supported by analysis of published p65 ChIP-seq data at the loci of these genes.

RELA REGULATORY POTENTIAL

MUC1-C interacts with TFs and other proteins that regulate gene expression; however, MUC1-C does not directly bind to DNA. As such, well-controlled MUC1-C ChIP-seq studies have been a challenge. To address involvement of MUC1-C in regulating the *SET1A*, *WDR5* and *RBBP5* genes, we have included data demonstrating that MUC1-C is necessary for their transcription (new Supplemental Fig. S2A). Consistent with our results, we have also found that RELA has a higher regulatory potential at *SET1A* and *WDR5*, as compared to *SET1B* and *ASH2L* at 1 kb distance from the TSS (CISTROME database <http://dbtoolkit.cistrome.org/>) (shown above). These findings will need to be confirmed and extended in subsequent studies, which we would respectfully propose are beyond the scope of the present work.

In Fig. 3 (p65 depletion), SET1B should be included along with SET1A. Additionally, along with comments 3 and 4, please show that ATAC-seq genome browser shot for SET1A (transcription downregulated upon MUC1-C depletion) vs. SET1B (transcription not regulated upon MUC1-C depletion). The authors could also analyze by ChIP-qPCR whether MUC1-C occupies SET1A vs. SET1B promoter and enhancers, revealing perhaps how gene selectivity could be achieved.

In response, we have included SET1B in Fig. 3A. In contrast to SET1A, the SET1B genome browser snapshot does not reflect significant MUC1-C regulation of chromatin accessibility. This data has been included in new Supplemental Fig. S3D.

6. Western blots show that depletion of MUC1-C reduces expression of SET1A, WDR5, RBBP5, several AP-1 members, but not of SET1B, ASH2L, Jun. Is this also seen in the RNA-seq data, are they part of the 100 downregulated genes in Fig. 4B/C? If yes, they could be highlighted in the Figure, if not, it should be discussed.

As confirmation of COMPASS gene regulation, analysis of nascent RNA synthesis demonstrated that MUC1-C drives transcription of SET1A, WDR5 and RBBP5, but not SET1B or ASH2L (new Supplemental Fig. S2A). Time-course analysis further confirmed that silencing MUC1-C decreases SET1A, and not SET1B, transcripts and protein (new Supplemental Figs. S1C and S1D). Downregulation of SET1A, WDR5 and RBBP5 expression was however not significant by analysis of the RNA-seq data using DESeq2. A statement to this effect has been included in the Results section.

7. The introduction should be improved to concisely summarize what is already known about the genomic actions of MUC1-C, and what are the rationale, key questions, hypotheses of the current study? A meaningful rationale would be, for example, to start with the RNA-seq analysis to identify COMPASS subunits (SET1A, WDR5) as hitherto unknown MUC1-C targets (control vs. depletion, high vs. low expression). A weaker rationale is the possible link between MUC1-C and H3K4me3: H3K4me3 is an established promoter (sometime also enhancer) activity mark, thus the altered expression of any transcription factor and cofactor that changes gene transcription will cause changes in H3K4me3 at promoters (enhancers) of regulated genes. It could be correlation, cause or consequence, and would in most cases not require any direct link of the (co-)factor to COMPASS.

As suggested, we have revised the last paragraph of the Introduction by summarizing the rationale for the current study and by including the identification of COMPASS subunits as MUC1-C targets.

REVIEWERS' COMMENTS:

Reviewer #2 (Remarks to the Author):

The revised manuscript includes answers to the comments. However, a few fundamental issues are unsolved, which are directly to do with the title.

1. NF-kappaB in the inflammatory, i.e., host cells is the central element of inflammation. However, NF-kappaB-driven processes in cancer cells have little to do with inflammation. Therefore, the argument in this paper is confusing.
2. Descriptions on the biology of TNBC is mixed up with stemness of cancer cells. This does not seem to be right and makes the argument confusing. The logic should be focused on CSC nature but not on TNBC biology.

Reviewer #3 (Remarks to the Author):

Most of my major comments have been satisfactory addressed in the revision.

I realize and accept that additional ChIP-seq data may not be possible for this study due to lack of high-quality antibodies and expertise in the lab. I would, however, strongly recommend to develop these techniques for follow-up studies.

The following editions of the revised Figures must be made:

In my PDF version of the revised manuscript including all revised Figures, it seems that the revised Figure 8 (A-E) is missing. Instead, the supplemental Figure 8 (A, B) has been inserted by mistake. . Please correct and check all Figures for completeness.

In the revised Figures 5D, 6E and 7A and in revised Supplemental Figure S7A, the position (bp distance from the transcription start site) of the NF-KB and AP-1 binding sites seems not to be labelled in the correct order, please edit.

September 1, 2023

Christina Karlsson-Rosenthal, PhD
Chief Editor
Silvia Belluti, PhD
Editorial Board Member
Communications Biology

RE: COMMSBIO-23-1325A

Dear Drs. Karlsson-Rosenthal and Belluti,

We appreciate the recent rereview of our manuscript "**MUC1-C INTERSECTS CHRONIC INFLAMMATION WITH EPIGENETIC REPROGRAMMING BY REGULATING THE SET1A COMPASS COMPLEX IN CANCER PROGRESSION**".

Comments of the reviewers and format requirements have been addressed in the following responses.

Reviewer #2

1. NF-kappaB in the inflammatory, i.e., host cells is the central element of inflammation. However, NF-kappaB-driven processes in cancer cells have little to do with inflammation. Therefore, the argument in this paper is confusing.

In response, we would respectfully argue that NF- κ B-driven signaling in cancer cells has been clearly linked to intrinsic chronic inflammation. A statement to this effect and reference to a supporting review (Taniguchi K and Karin M, *NF- κ B, inflammation, immunity and cancer, coming of age*, Nature Rev. Immunol., 2018) have been included in the Introduction.

2. Descriptions on the biology of TNBC is mixed up with stemness of cancer cells. This does not seem to be right and makes the argument confusing. The logic should be focused on CSC nature but not on TNBC biology.

In the previous critique, Reviewer #2 commented that: "One of the most important points to be included in the revised paper would be how this process is related to anything unique to the biology of TNBC?" Accordingly, we revised the Introduction with a focus on TNBC biology.

Here, Reviewer #2 now suggests that the focus should be on CSCs and not on TNBC biology.

In response, we have modified the Introduction by stating that "The present studies focus on the involvement of MUC1-C in integrating the activation of chronic inflammation with epigenetic reprogramming as a mechanism that contributes to the CSC state and the pathogenesis TNBCs and potentially other cancers."

Reviewer #3

1. In my PDF version of the revised manuscript including all revised Figures, it seems that the revised Figure 8 (A-E) is missing. Instead, the supplemental Figure 8 (A, B) has been inserted by mistake. Please correct and check all Figures for completeness.

As suggested, we have corrected and checked the Figures for completeness.

2. In the revised Figures 5D, 6E and 7A and in revised Supplemental Figure S7A, the position (bp distance from the transcription start site) of the NF- κ B and AP-1 binding sites seems not to be labelled in the correct order, please edit.

As requested, we have edited positioning of the NF- κ B and AP-1 binding sites in the above mentioned figures.

Format Requirements

A completed version of the Final Revision Instructions file has been uploaded as a Related Manuscript.

We have (i) formatted the figure numbers in lower case, (ii) included the unedited and uncropped gel images as supplemental figures, and (iii) compiled excel files for the main figures to provide access to the numerical source data for graphs and charts.

We are submitting clean and highlighted versions of the revised manuscript in response to the above comments.

The reconsideration of this work is appreciated.

Sincerely,

Donald W. Kufe, MD